# Dynamic Loss-Based Sample Reweighting for Improved Large Language Model Pretraining

**Daouda A. Sow**\*
The Ohio State University
sow.53@osu.edu

**Herbert Woisetschläger**\*
Technical University of Munich
h.woisetschlaeger@tum.de

**Saikiran Bulusu**
The Ohio State University
bulusu.11@osu.edu

**Shiqiang Wang**
IBM Research
wangshiq@us.ibm.com

**Hans-Arno Jacobsen**
University of Toronto
jacobsen@eecg.toronto.edu

**Yingbin Liang**
The Ohio State University
liang889@osu.edu

## Abstract

Pretraining large language models (LLMs) on vast and heterogeneous datasets is crucial for achieving state-of-the-art performance across diverse downstream tasks. However, current training paradigms treat all samples equally, overlooking the importance or relevance of individual samples throughout the training process. Existing reweighting strategies, which primarily focus on group-level data importance, fail to leverage fine-grained instance-level information and do not adapt dynamically to individual sample importance as training progresses. In this paper, we introduce novel algorithms for dynamic, instance-level data reweighting aimed at improving both the efficiency and effectiveness of LLM pretraining. Our methods adjust the weight of each training sample based on its loss value in an online fashion, allowing the model to dynamically focus on more informative or important samples at the current training stage. In particular, our framework allows us to systematically devise reweighting strategies deprioritizing redundant or uninformative data, which we find tend to work best. Furthermore, we develop a new theoretical framework for analyzing the impact of loss-based reweighting on the convergence of gradient-based optimization, providing the first formal characterization of how these strategies affect convergence bounds. We empirically validate our approach across a spectrum of tasks, from pretraining 7B and 1.4B parameter LLMs to smaller-scale language models and linear regression problems, demonstrating that our loss-based reweighting approach can lead to faster convergence and significantly improved performance.

## 1 Introduction

The rapid advancement of large language models (LLMs) (Brown et al., 2020; Raffel et al., 2020; Touvron et al., 2023; Chowdhery et al., 2023; Achiam et al., 2023; Dubey et al., 2024) has revolutionized natural language processing and artificial intelligence (AI) capabilities. These models, trained on billions or trillions of tokens, exhibit remarkable generalization capabilities across a wide range of downstream tasks. As datasets grow to web-scale proportions and models become increasingly large, the need for more efficient training techniques has become paramount.

Existing approaches to LLM pretraining predominantly involve two phases: heavy data curation (Longpre et al., 2023; Dubey et al., 2024; Wettig et al., 2024; Penedo et al., 2024) and training with uniform sampling on the constructed corpus. Data curation for LLM pretraining typically involves a combination of automated filtering techniques and manual quality checks. For instance, heuristic-based filters are often employed to remove low-quality content and deduplicate data. Some approaches use perplexity-based filtering or auxiliary classifiers (Gao et al., 2020; Penedo et al., 2023) to identify high-quality samples. Manual curation is then applied to refine these filtered datasets, often involving human evaluation of subsets of the data to ensure quality and relevance. However,

---

\*Daouda Sow and Herbert Woisetschläger were interns at IBM Research when most of this work was done.

these approaches face significant limitations due to scalability issues and the static nature of data selection. First, as the pretraining corpora grow to hundreds of billions of tokens, manual curation becomes increasingly infeasible. The sheer volume of data makes it impractical for humans to review even a small fraction of the dataset. Second, data curation methods cannot adapt to the changing importance of samples during the training process, e.g., as the model improves over time certain samples that were useful early in training can become irrelevant in later training stages.

In addition, conventional gradient-based training pipelines often treat all data points as equally informative regardless of their individual importance at current training stage. This uniform sampling strategy, while simple, overlooks the nuanced diversity within large-scale datasets and potentially wastes computational resources on less informative samples. Recent work (Xie et al., 2023; Fan et al., 2023) has explored group-level reweighting strategies, which adjust the importance of entire domains or groups of data. While these methods have shown promise, they operate at a coarse level, do not fully leverage the fine-grained information available at the instance level, and still lack dynamicity in importance assignment during the training phase.

Given these limitations, this paper seeks to address the following fundamental question:

*How can we dynamically leverage instance-level information to accelerate training and improve model performance without incurring significant computational overhead while potentially reducing the need for extensive data curation?*

Answering this question presents several intertwined technical challenges, particularly in the context of LLM pretraining. First, unlike in computer vision, where samples are seen repeatedly over the course of training (often hundreds of times), in LLM pretraining, individual samples are typically encountered only once due to the vast size of the datasets. This renders dynamic reweighting approaches primarily devised for multi-epoch training or relying on historical statistics, such as previous loss or gradient norms (Loshchilov & Hutter, 2015; Jiang et al., 2019), inadequate within the context of LLM pretraining. Second, storing previous statistics for individual samples becomes computationally infeasible when working with web-scale datasets, and methods requiring the persistent storage of sample-level data would significantly inflate resource requirements. Furthermore, for a general-purpose LLM, leveraging small hold-out validation sets to compute importance weights through, e.g., bilevel optimization as in Grangier et al. (2023), is ineffective.

Addressing these challenges, our work makes the following significant contributions:

**Instance-Level Loss-Based Reweighting Strategies.** We introduce and systematically study a variety of instance-level, loss-based reweighting strategies for improving both the efficiency and effectiveness of ML training, especially within the context of LLM pretraining. Each strategy is meticulously designed to achieve specific goals, such as focusing the learning dynamics on different parts of the loss distribution. This builds on recent work that emphasizes the importance of data diversity and sample importance in pretraining and extends previous works on group-level reweighting for LLM pretraining, such as DoReMi Xie et al. (2023), DoGE Fan et al. (2023). It also adds a new dimension by incorporating more fine-grained per-sample dynamics rather than domain-level adjustments. In fact, combining our reweighting schemes with DoGE/DoReMi achieves better or comparable performance on various few-shot reasoning benchmarks when we train on the SlimPajama dataset. Moreover, our study reveals that, in general, strategies down-weighting the importance of low-loss samples tend to consistently yield performance improvements across various scenarios.

**New Theoretical Framework.** We develop a new theoretical framework for analyzing the effects of loss-based reweighting on training acceleration. To the best of our knowledge, this represents the first explicit characterization of loss reweighting effects within the convergence bounds of gradient methods under widely adopted loss geometries. Our derived convergence bounds provide theoretical justification for the empirical success of downweighting low-loss samples, demonstrating faster convergence under this strategy.

**Empirical Validation.** We conduct extensive experiments that corroborate our claims and support our theoretical findings. Notably, we demonstrate that the advantages of downweighting low-loss samples are observed across a spectrum of problem scales and complexities: (i) in complex, large-scale problems such as LLM pretraining with number of parameters ranging from 124M to 7B, our approach leads to notably improved performance and faster convergence; (ii) in simple, small-scale problems like linear regression, the results highlight the fundamental nature of our findings.

## 2 RELATED WORK

**Training Data Re-weighting/Selection for LLMs.** Several recent studies (Xie et al., 2023; Chen et al., 2023; Fan et al., 2023; Thakkar et al., 2023) have explored various reweighting techniques to enhance the generalization and efficiency of language models pretraining. For instance, Xie et al. (2023) and Fan et al. (2023) optimize the composition of pretraining corpora to achieve better performance across pretraining domains or for out-of-domain generalization. Chen et al. (2023) introduce a framework for ordered skill learning, optimizing data selection based on how effectively it teaches interdependent skills for continual pretraining and fine-tuning regimes. Although effective, these techniques operate at the group level, whereas our work explores reweighting at the instance level, offering finer control over how individual samples are treated based on their loss values. Furthermore, we demonstrate that combining domain-level methods such as DoReMi (Xie et al., 2023) or DoGE (Fan et al., 2023) with our instance-level reweighting methods results in improved performance across multiple domains. Instance-level reweighting has been used in post-training settings of LLMs (Chen et al., 2024; Jiang et al., 2024). Jiang et al. (2024) boost the self-improvement abilities of LLMs by employing sample reweighting to filter out self-generated data that have correct answers but exhibit high distribution shifts. Chen et al. (2024) reweight individual samples during continual training/instruction-tuning to focus on medium-loss samples. In contrast, our work systematically studies the effects of various sample-level, loss-based reweighting strategies on the efficiency and effectiveness of LLMs *pretraining*. The approach in Fan & Jaggi (2023) offers a curriculum learning framework that prioritizes samples with a higher learnability score, which is precomputed using another auxiliary model similar to DoReMi and DoGE. While we do not explicitly address curriculum learning in this work, our re-weighting mechanisms naturally allow for implementing a form of loss-based curriculum learning algorithms without the need to train and store additional proxy models as in Fan & Jaggi (2023).

**Sample-level Re-weighting as generic ML solution.** Sample-level reweighting has been extensively explored in other machine learning areas. For image classification, Loshchilov & Hutter (2015), Jiang et al. (2019), and Katharopoulos & Fleuret (2018) introduce pioneering approaches that prioritize samples based on loss values or gradient norms with the goal of accelerating training speed. Although these methods emphasize the importance of selecting high-loss samples during training, they typically require additional forward/backward passes on each training sample and are primarily designed for multi-epoch training, limiting their applicability with the vast pretraining corpus used for LLMs. Using bilevel optimization, Grangier et al. (2023) adapt the data distribution during training to focus more on relevant samples for a target data distribution. Similarly, Ren et al. (2018) employ a meta-learning approach to adjust sample weights based on validation set performance, which excels in noisy and imbalanced datasets. In this work, we introduce and investigate various lightweight, loss-based reweighting techniques that add little to no computational overhead compared to uniform sampling and do not require any nested optimization routines, often arising with bilevel optimization and/or meta-learning. Sample-level reweighting has also been explored in the context of adversarial machine learning (Zhang et al., 2020; Liu et al., 2021; Zeng et al., 2021; Sow et al., 2023), domain adaptation (Jiang & Zhai, 2007; Fang et al., 2020), data augmentation (Yi et al., 2021), and imbalanced classification (Qi et al., 2021; Ren et al., 2018). Our reweighting mechanisms also have the potential to be studied under these contexts, which we leave as future work.

## 3 PRELIMINARIES: AUTOREGRESSIVE LANGUAGE MODELING & GOAL

Large language models (LLMs) are typically trained using autoregressive language modeling, where the objective is to predict the next token in a sequence given the previous tokens. Formally, given a sample sequence of tokens $\mathbf{x} = (x_1, x_2, \ldots, x_T)$, where $x_t$ represents the $t$-th token in the sequence, the LLM model parameterized by $\theta \in \mathbb{R}^d$ is trained to maximize the likelihood of the sequence:

$$P_{\text{LLM}}(\mathbf{x}; \theta) = \prod_{t=1}^{T} P(x_t | x_1, \ldots, x_{t-1}; \theta).$$

where $P(x_t | x_1, \ldots, x_{t-1}; \theta)$ denotes the conditional probability of generating token $x_t$ given $\theta$ and all previously seen tokens $x_1, \ldots, x_{t-1}$. In practice, autoregressive LLMs typically compute the loss $f(\mathbf{x}; \theta)$ for sample $\mathbf{x}$ using the negative log-likelihood (NLL) of the predicted tokens.

$$f(\mathbf{x}; \theta) = -\log P_{\text{LLM}}(\mathbf{x}; \theta) = -\sum_{t=1}^{T} \log P(x_t | x_1, \ldots, x_{t-1}; \theta).$$

Based on these sample losses, conventional SGD-like algorithms will then update the model parameters with the average gradient over all samples in a batch $\mathcal{B}$ at each training step $t$, i.e.,

$$\theta^{t+1} = \theta^t - \frac{\eta}{|\mathcal{B}|} \sum_{i \in \mathcal{B}} \nabla f(\mathbf{x}_i; \theta^t), \tag{1}$$

which results in each sample contributing equally to the overall training update. While this approach enjoys simplicity, it inherently treats all data points as equally informative, regardless of their individual relevance or difficulty. This uniform treatment overlooks the nuanced diversity within the data, which is particularly critical for current LLMs trained on vast and heterogeneous web-scale datasets. In such scenarios, the ability to dynamically differentiate between high-value and lesser-value data samples at different training stages is paramount for optimizing training efficiency and model performance.

In contrast, our work introduces and benchmarks a variety of simple, lightweight reweighting strategies that can be seamlessly integrated into existing training pipelines with little to no computational overhead. These strategies aim to address the inherent limitations of the standard uniform averaging method by assigning dynamic importance to individual data samples based on their loss values, thus providing a more refined control over the training process. Ultimately, our objective is twofold: (i) to improve the efficiency and effectiveness of LLM training by allowing the model to dynamically focus on more relevant or informative data at different training stages and (ii) to potentially provide a scalable, automated approach to online data selection during pretraining. This ambitiously seeks to reduce the need for extensive manual data curation and selection efforts, which are increasingly becoming infeasible given the growing size and diversity of pretraining corpora. We envision these reweighting strategies as a foundation for more efficient and adaptive data utilization in LLM pretraining, ultimately reducing the time, cost, and complexity associated with model pretraining.

## 4 APPROACH

### 4.1 TECHNICAL CHALLENGE: DATA REWEIGHTING FOR LLMs MUST BE DYNAMIC AND FULLY ONLINE

The core of our framework is to *dynamically* reweight individual training samples based on their importance at different training stages. Although sample importance seems static—intuitively, "garbage data is garbage data" regardless of model or training stage—data reweighting should be dynamic. For instance, similar or duplicated samples in training corpora may be useful early in training, but should be deprioritized once the model captures their patterns. Similarly, a hard but useful sample could be assigned a larger weight, however, as the model learns it could be beneficial to reduce its weight. Based on these observations, we update the model at each step $t$ using

$$\theta^{t+1} = \theta^t - \eta_t \sum_{i \in \mathcal{B}} w(\mathbf{x}_i; \theta^t) \nabla f(\mathbf{x}_i; \theta^t), \tag{2}$$

where $\sum_{i \in \mathcal{B}} w(\mathbf{x}_i; \theta^t) = 1$ and the weight $w(\mathbf{x}_i; \theta^t)$ is dynamically assigned based on sample $\mathbf{x}_i$ and current model $\theta^t$. However, designing effective dynamic reweighting methods for LLM pretraining involves several inherent complexities. In LLM pretraining, each sample is typically seen only once due to the vast size of the datasets, which contrasts with problems where repeated exposure to data is common, such as in computer vision problems. This makes methods that depend on accumulating historical data, such as previous loss or gradient norms (Loshchilov & Hutter, 2015; Jiang et al., 2019), unsuitable. Furthermore, the sheer scale of these large corpus makes it computationally prohibitive to track and store sample-specific statistics, as it would significantly increase resource requirements. Additionally, approaches that rely on hold-out validation sets to compute sample weights, as in Grangier et al. (2023), are also irrelevant for pretraining a general-purpose LLM. These unique technical challenges necessitate the development of a fully online, lightweight reweighting approach that scales efficiently with the size of a modern LLM training corpus, which we discuss next.

### 4.2 LOSS-BASED REWEIGHTING STRATEGIES

To address the aforementioned challenges, we propose a novel framework of loss-based sample reweighting strategies motivated by the following theorem. Theorem 1 characterizes the effects

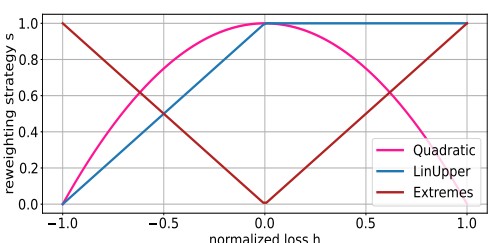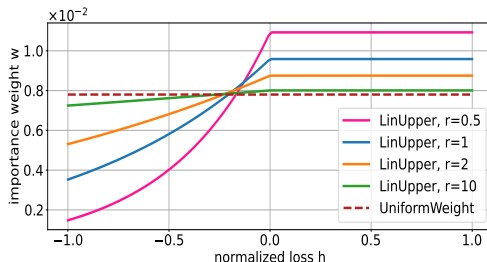

Figure 1: **Left:** Geometric curves of the different reweighting functions. **Right:** Shape of the `LinUpper` strategy after applying Eq. (4) on top of it for different values of $r$. These plots are obtained for a batch of 128 uniformly drawn losses. As $r$ increases, `LinUpper` converges to the uniform averaging method.

of loss-based reweighting on the convergence bound of reweighted full gradient descent when the sample losses $f(\mathbf{x}_i; \theta^t)$ are convex. The extension to the stochastic setting and analysis for other types of loss geometries is deferred to Section 5. Our full statement of Theorem 1 and detailed proofs for all theoretical statements can be found in the appendix.

**Theorem 1.** *Consider $M$ data points and let each loss $f(\mathbf{x}_i; \cdot)$ be convex. Further, assume the interpolation regime holds, i.e., $\exists \theta^* \in \mathbb{R}^d$ such that $\theta^* \in \arg\min_{\theta \in \mathbb{R}^d} f(\mathbf{x}_i; \theta) \; \forall i$. Then, for a reweighting scheme that satisfies $\max_{i \in \{1,\ldots,M\}} w(\mathbf{x}_i; \theta^t) \leq 2/M$, we have*

$$\frac{1}{M}\sum_{i=1}^{M} f(\mathbf{x}_i; \bar{\theta}^T) - \frac{1}{M}\sum_{i=1}^{M} f(\mathbf{x}_i; \theta^*) \leq \mathcal{O}\left(\frac{\|\theta^0 - \theta^*\|^2}{T}\right) + \frac{1}{T}\sum_{t=0}^{T-1}\delta^t, \tag{3}$$

*where $\delta^t = \sum_{i=1}^{M}\left(\frac{1}{M} - w(\mathbf{x}_i; \theta^t)\right)(f(\mathbf{x}_i; \theta^t) - f(\mathbf{x}_i; \theta^*))$ and $\bar{\theta}^T = \frac{1}{T}\sum_{t=0}^{T-1}\theta^t$.*

Theorem 1 illustrates how setting the importance weight $w(\mathbf{x}_i; \theta^t)$ based on the loss gap $f(\mathbf{x}_i; \theta^t) - f(\mathbf{x}_i; \theta^*)$ influences the convergence bound. Specifically, we make the following key observations: (i) fixing $w(\mathbf{x}_i; \theta^t) = \frac{1}{M} \; \forall i$, leads to $\delta_t = 0$ and we recover the traditional convergence bound of gradient descent for convex functions; (ii) assigning larger weights to the smaller loss gaps leads to $\delta_t \geq 0$, which yields a looser convergence bound; (iii) to achieve a better bound ($\delta_t \leq 0$), one should down-weight the importance of samples with low loss gaps. Further, note that Eq. (3) critically holds only for reweighting schemes satisfying $w(\mathbf{x}_i; \theta^t) \leq 2/M$, which puts an upper bound on the maximum weight that can be assigned to a single data point, and hence, eliminates worst-case importance assignment strategies—such as those based on distributionally robust optimization (Qian et al., 2019; Qi et al., 2021; Kumar et al., 2023)—which are prone to overfitting to outliers, a particularly problematic issue when dealing with the large-scale, noisy corpora used in LLM pretraining. Finally, we also note that $\max_{i \in \{1,\ldots,M\}} w(\mathbf{x}_i; \theta^t) \leq 2/M$ essentially means that the weight of any sample (after reweighting) should not be more than twice the uniform weight. This upper bound on the sample weights is directly captured by our convergence analysis. Please see Appendix A.5 for discussions on how it can be satisfied by our proposed reweighting methods.

Based on the insights from Theorem 1 on the relationship between loss and importance weight, we study a series of reweighting strategies, each carefully designed to focus the learning dynamics on different parts of the loss distribution during training. Let $w_i$ be the importance weight for sample $\mathbf{x}_i$ where we drop the dependency on $\theta^t$ for clarity. We first normalize the individual sample losses into a bounded range $[-\alpha, \alpha]$. This step crucially ensures the loss values are scaled consistently and amenable to the subsequent weighting functions. We then apply the following analytical functions on the normalized loss $h_i$. Figure 1 shows the geometric curves of these different functions.

- **Linear Upper-Bound Strategy (LinUpper).** In this strategy, the weight is proportional to the normalized loss but is capped at a predefined $\alpha$ value, ensuring that outliers do not dominate the training process. The functional form is $s_i := \min\{h_i + \alpha, \alpha\}$.

- **Quadratic Strategy (Quadratic).** This strategy prioritizes samples with moderate loss values while down-weighting both low-loss (easy or repetitive) and high-loss (potential outliers) samples by applying a quadratic function $s_i := \alpha\left(1 - \frac{h_i^2}{\alpha^2}\right)$.

- **Extremes-Based Strategy (Extremes).** This scheme emphasizes both the hardest (high-loss) and the easiest (low-loss) samples by applying $s_i := |h_i|$, ensuring that samples at the extremes of the loss distribution receive higher importance weights.

To further enhance training stability and effectiveness, we add a curriculum-based adjustment mechanism on top of the strategic sample weight $s_i$ by controlling the sharpness of the importance weighting using a temperature parameter $r$. The final weight $w_i$ is computed as:

$$w_i = \frac{e^{s_i/r}}{\sum_j e^{s_j/r}}, \tag{4}$$

where $r$ is annealed during training. Early in training, when the model is still learning fundamental patterns and losses are less informative, we employ a more uniform weighting scheme to ensure diverse feature learning by adopting a large $r$ (see Figure 1). As training progresses, we decrease the value of $r$ to enact full use of our reweighting strategies. Algorithm 1 depicts our approach. It is important to note that our reweighting approach adds nearly zero computational overhead. During the forward pass, we already compute the sample losses, which are then used to derive the importance weights through simple analytical transformations. Since no extra loss or gradient computations are required, our reweighting strategies can be seamlessly integrated into standard training pipelines without incurring significant additional costs in terms of time or resources.

---

**Algorithm 1** Fully Online Instance Reweighting

---

1: **Input:** stepsize $\eta$, initial model parameters $\theta_0 \in \mathbb{R}^d$, temperature $\{r_t\}_{t=0}^T$, $f_{\min}$ and $f_{\max}$.
2: **for** $t = 0, 1, 2, ..., T-1$ **do**
3:     Draw a minibatch of data samples $\mathcal{B} = \{\mathbf{x}_i\}_{i=1}^b$
4:     Run forward pass to compute losses $\{f_{i,t}\}_{i=1}^b$ for all samples in $\mathcal{B}$
5:     Normalize losses into interval $[-\alpha, \alpha]$ using linear transform: $h_{i,t} = \frac{2(f_{i,t} - f_{\min})}{f_{\max} - f_{\min}} - 1$
6:     Apply reweighting strategy to normalized losses to obtain $\{s_{i,t}\}_{i=1}^b$ // {LinUpper, Quadratic, Extremes}
7:     Apply curriculum adjustment using Eq. (4) to obtain sample weights $\{w_{i,t}\}_{i=1}^b$
8:     Update $\theta^{t+1} = \theta^t - \eta \sum_{i \in \mathcal{B}} w_{i,t} \nabla f_{i,t}$
9: **end for**

---

**Optimal strategy minimizing $\delta^t$.** We next demonstrate that applying Eq. (4) on top of our strategy `LinUpper`, in fact, corresponds to the optimal strategy that minimizes a KL-divergence regularized version of $\delta^t$ in Theorem 1.

**Proposition 1.** *The optimal strategy that minimizes the KL-divergence regularized $\delta^t$, i.e., $\delta^t + r \sum_{i=1}^M w_i \log(M w_i)$, is given by*

$$w_i = C \min \left\{ \exp\left(\frac{h_i}{r}\right), \frac{2}{M} \right\}, \tag{5}$$

*where $C$ is a normalizing constant that ensures $\sum_i w_i = 1$.*

*Proof.* Proof relegated to Appendix C.2. □

As shown in Theorem 1 and Proposition 1, our experiments confirm that the `LinUpper` reweighting scheme indeed achieves faster convergence and improved performance across a spectrum of LLMs pretraining scales. The incorporation of KL-divergence regularization in the optimization of $\delta^t$ prevents extreme solutions that focus only on the highest-loss samples. Without this regularization term, the optimal solution becomes trivial: it assigns a weight of $\frac{2}{M}$ to the $\frac{M}{2}$ samples with the highest losses and zero weight to the remaining samples. It is important to note that such a strategy discards half of the dataset and performs poorly in practice. However, our `LinUpper` method offers a balance between focusing on high-loss samples and maintaining the diversity of data utilized during training, leading to faster convergence and performance improvements.

## 5   Analysis of Loss-based Reweighting on Convergence Bound

In this section, we analyze the effects of loss-based reweighting on the convergence bound in the more generic setting of stochastic gradient methods possibly with momentum. Consider the problem

of minimizing a finite sum of $M$ objective functions:

$$\min_{\theta \in \mathbb{R}^d} \left\{ f(\theta) := \frac{1}{M} \sum_{i=1}^{M} f_i(\theta) \right\} \tag{6}$$

where each $f_i : \mathbb{R}^d \to \mathbb{R}$ corresponds to a sample loss function $f(\mathbf{x}_i; \cdot)$. We make the following widely adopted assumptions:

**Assumption 1** (Convexity). *Each function $f_i$ is convex, i.e., for all $\theta, \theta' \in \mathbb{R}^d$ and $i \in 1, \ldots, M$:*

$$f_i(\theta') \geq f_i(\theta) + \langle \nabla f_i(\theta), \theta' - \theta \rangle. \tag{7}$$

**Assumption 2** (*L*-smoothness). *Each function $f_i$ is $L_i$-smooth. Formally, for all $\theta, \theta' \in \mathbb{R}^d$ and $i \in 1, \ldots, M$:*

$$\left\| \nabla f_i(\theta') - \nabla f_i(\theta) \right\| \leq L_i \left\| \theta' - \theta \right\|, \tag{8}$$

*and $L = \max_{i \in [M]} L_i$.*

**Assumption 3** (Interpolation condition). *There exists a global optimum $\theta^* \in \mathbb{R}^d$ such that:*

$$\theta^* \in \arg \min_{\theta \in \mathbb{R}^d} f_i(\theta), \quad \forall i \in \{1, \ldots, M\}. \tag{9}$$

Assumption 3 is widely adopted in modern ML settings (Dar et al., 2021; Radhakrishnan et al., 2020; Jhunjhunwala et al., 2023), particularly when analyzing overparameterized deep neural networks. This ensures that all individual sample objectives can be minimized simultaneously, effectively eliminating adversarial trade-offs where minimizing one objective could worsen another.

Next, we characterize the effects of loss-based reweighting on the convergence bound of reweighted minibatch SGD when the sample losses $f_i$ are convex. For ease of notation, we denote $w(\mathbf{x}_i; \theta^t)$ by $w_{i,t}$ in this section and the corresponding proofs in the Appendix.

**Theorem 2** (Minibatch SGD with momentum). *Let Assumptions 1, 2, and 3 hold. Consider a minibatch of size $|\mathcal{B}| = b$ with reweighting scheme satisfying $\max_{i \in \mathcal{B}} w_{i,t} \leq 2/b$. Then, for $\eta = \frac{1}{8L\sqrt{T+1}}, \lambda_t > 0$, we have*

$$\mathbb{E}[f(\theta^T) - f(\theta^*)] \leq \frac{8L\left\| \theta^0 - \theta^* \right\|^2}{\sqrt{T+1}} + \frac{2}{T+1} \sum_{t=0}^{T-1} \delta_t + \frac{2}{T+1} \sum_{t=0}^{T-1} \lambda_t \mu_t, \tag{10}$$

*where $\delta_t = \mathbb{E}\left[ \sum_{i \in \mathcal{B}} \left( \frac{1}{b} - w_{i,t} \right) (f_i(\theta^t) - f_i(\theta^*)) \right]$, $\mu_t = \mathbb{E}\left[ \sum_{i \in \mathcal{B}} \left( \frac{1}{b} - w_{i,t} \right) (f_i(\theta^t) - f_i(\theta^{t-1})) \right]$, and $\bar{\theta}^T = \frac{1}{T} \sum_{t=0}^{T-1} \theta^t$.*

*Proof.* Proof provided in Appendix C.3. $\qquad \square$

We note similar observations as in Theorem 1 and again down-weighting the importance of samples with low loss gaps leads to a better bound. Typically, the minibatch gradient with uniform weights is unbiased, $\mathbb{E}\left[ \sum_{i \in \mathcal{B}} \frac{1}{b} \nabla f_i(\theta^t) \right] = \nabla f(\theta^t)$. However, in our setting, the importance weights are functions of the sample functions, hence, leading to biased gradients and the analysis of Theorem 2 being more involved. Further, we provide the analysis for minibatch SGD when the sample functions are non-convex in Appendix C.4.

# 6 EXPERIMENTAL EVALUATION

## 6.1 TRAINING AND EVALUATION SETUP

We conduct our first set of experiments using decoder-only transformer models (Vaswani, 2017; Radford et al., 2019) with parameter sizes of 120M, 210M, and 300M, which we refer to as, respectively, GPT2-mini, GPT2-small, and GPT2-medium. We train on the SlimPajama (Soboleva et al., 2023) corpus that includes seven diverse domains: Common Crawl (CC), C4, GitHub, StackExchange, Book, Arxiv, and Wikipedia. In the first step, we pretrain the three GPT2 models on all seven domains where we compare our sample-level reweighting methods (`LinUpper`, `Quadratic`, `Extremes`) against the uniform averaging baseline in which each sample contributes equally. In a second step, we assess the impact of combining our reweighting techniques with existing domain-level reweighting

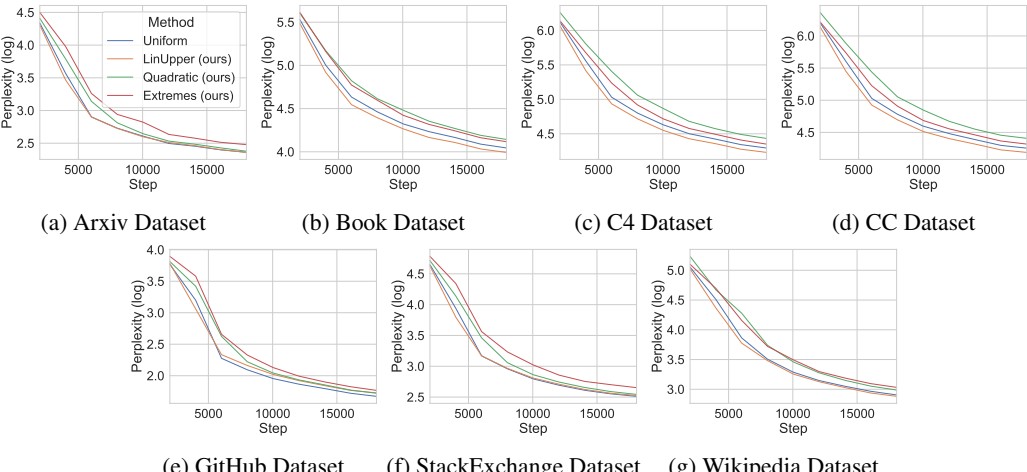

(a) Arxiv Dataset     (b) Book Dataset     (c) C4 Dataset     (d) CC Dataset

(e) GitHub Dataset     (f) StackExchange Dataset     (g) Wikipedia Dataset

Figure 2: Per-domain perplexities on hold-out validation sets under the uniform domain sampling setting for the GPT2-medium model. Our reweighting strategy `LinUpper` strategy achieves better or at least comparable perplexity on 5 out of 7 domains.

methods, specifically DoGE and DoReMi, to demonstrate the potential for further performance gains when fine-grained sample reweighting is added on top of domain-level adjustments. Then, we conduct experiments of pretraining larger models with 1.4B and 7B parameters (Section 6.4). We also provide results for a regression problem on synthetic data in Appendix B, to validate the fundamental advantage of reweighting. Model architecture details can be found in Appendix A.2. Our codebase for the GPT2 experiments is publicly available.[1]

We measure performance using two types of metrics:

**Perplexity on Training Distribution.** We evaluate per-domain perplexity and average perplexity on all training domains using hold-out validation sets. By doing so, we aim to generate insights on the sample efficiency during pretraining, i.e., how well a model can learn from a given dataset and adapt to the data distribution.

**Few-Shot Reasoning Ability.** To assess generalization beyond the pretraining distribution, we evaluate models on a series of diverse reasoning tasks in a 5-shot setting, including LogiQA Liu et al. (2020), LogiQA 2 Liu et al. (2023), SciQ Welbl et al. (2017), and PiQA Bisk et al. (2020).

## 6.2 COMPARISONS UNDER UNIFORM DOMAIN SAMPLING

**Comparisons on Pretrainining Distribution.** Figure 2 shows the per-domain perplexity for the GPT2-medium model on all seven domains. We provide the final per-domain perplexity results for the GPT2-mini and GPT2-small models in Table 1. The perplexity plots for GPT2-mini and GPT2-small can be found in Appendix A.3. Figure 2 shows that our proposed `LinUpper` strategy outperforms or matches the other baseline in 5 out of 7 domains, with particularly notable improvements in CC, C4, and Book domains. `LinUpper` achieves significantly lower perplexity in these domains, demonstrating its effectiveness at handling noisier data sources. This suggests that down-weighting low-loss samples –central to `LinUpper`– helps reduce the effects of redundant or similar data, and thus accelerates convergence as supported by our theoretical findings. We also note competitive results for `LinUpper` method at the smaller scale models in Table 1, but the advantage is more significant in the larger GPT2-medium model.

**Few-Shot Evaluations.** Table 2 provides the 5-shot evaluations on 4 different tasks for the GPT2-medium model. The `LinUpper` reweighting strategy consistently yields competitive results, outperforming the other compared baselines on 3 out of 4 tasks. These results further validate the general effectiveness of `LinUpper` across different types of reasoning challenges, including logical inference (LogiQA), physical commonsense reasoning (PiQA), and scientific reasoning (SciQ). Notably, `Quadratic` performs best in LogiQA 2 (28.6%) , indicating that reducing the effects of

---

[1]https://github.com/sowmaster/Sample-Level-Loss-Reweighting-ICLR-2025.

Table 1: Per-domain perplexities on hold-out validation datasets under the *uniform* domain sampling setting for the GPT2 models. Across a wide variety of domains, our `LinUpper` method outperforms the uniform baseline.

| Model | Method | Arxiv | Book | C4 | CC | Benchmark GitHub | StackExchange | Wikipedia | *Mean* |
|---|---|---|---|---|---|---|---|---|---|
| GPT2-mini | Uniform | **2.48** | 4.21 | 4.47 | 4.44 | **1.84** | **2.66** | 3.14 | 3.32 |
| | LinUpper (ours) | 2.49 | **4.16** | **4.41** | **4.38** | 1.89 | 2.67 | **3.12** | **3.30** |
| | Quadratic (ours) | **2.48** | 4.26 | 4.55 | 4.53 | 1.87 | 2.67 | 3.17 | 3.36 |
| | Extremes (ours) | 2.54 | 4.23 | 4.47 | 4.44 | 1.90 | 2.74 | 3.19 | 3.36 |
| GPT2-small | Uniform | **2.36** | 4.04 | 4.29 | 4.26 | **1.67** | **2.51** | 2.90 | 3.15 |
| | LinUpper (ours) | **2.36** | **3.99** | **4.23** | **4.19** | 1.73 | 2.53 | **2.88** | **3.13** |
| | Quadratic (ours) | 2.38 | 4.14 | 4.43 | 4.41 | 1.72 | 2.54 | 2.99 | 3.23 |
| | Extremes (ours) | 2.48 | 4.12 | 4.35 | 4.32 | 1.77 | 2.65 | 3.03 | 3.24 |
| GPT2-medium | Uniform | **2.34** | 4.02 | 4.27 | 4.22 | **1.66** | **2.49** | 2.88 | 3.13 |
| | LinUpper (ours) | **2.34** | **3.96** | **4.20** | **4.15** | 1.72 | 2.50 | **2.85** | **3.10** |
| | Quadratic (ours) | 2.35 | 4.11 | 4.40 | 4.36 | 1.70 | 2.51 | 2.93 | 3.19 |
| | Extremes (ours) | 2.45 | 4.08 | 4.31 | 4.27 | 1.76 | 2.62 | 3.00 | 3.21 |

Table 2: 5-shot Performance evaluations under *uniform* domain sampling for the GPT2-medium model. We report the accuracy for every benchmark. The highest accuracy per benchmark is highlighted in **bold**. The `LinUpper` reweighting strategy consistently yields competitive results, outperforming the other compared baselines on 3 out of 4 tasks.

| Benchmark Accuracy | Uniform | Method LinUpper (ours) | Quadratic (ours) | Extremes (ours) |
|---|---|---|---|---|
| LogiQA | 25.7 | **27.9** | 25.5 | 26.5 |
| LogiQA 2 | 27.5 | 27.6 | **28.6** | 27.7 |
| SciQ | 49.0 | **51.8** | 48.7 | 49.6 |
| PiQA | 55.5 | **56.2** | 54.6 | 55.2 |
| *Mean* | 39.4 | **40.9** | 39.3 | 39.8 |

both high-loss (potential outliers) and low-loss (potentially redundant data) samples can sometimes improve generalization performance even though it generally does not achieve best efficiency for learning the pretraining distribution. This suggests that the `Quadratic` scheme could be useful when training under noisier and less curated datasets. Also note that the `Extremes` strategy consistently lags behind the other methods, which further confirms the advantage of reducing the importance of low-loss samples.

## 6.3 COMPARISONS UNDER NON-UNIFORM DOMAIN SAMPLING

The performance of the compared methods under non-uniform domain sampling setting are presented in the Table 3. The results clearly demonstrate the value of incorporating our `LinUpper` strategy on top of existing domain-level reweighting methods such as DoGE and DoReMi. Across all tasks, `LinUpper` consistently improves the performance of both domain reweighting baselines, reinforcing the effectiveness of dynamically adjusting sample weights at the instance level in conjunction with domain-level reweighting. In particular, our reweighting strategy `LinUpper` provides notable improvements for LogiQA task, boosting accuracy from 27.2% to 28.6% with DoGE and from 27.2% to 27.6% with DoReMi. The gains in LogiQA 2 are similar, where our reweighting method `LinUpper` consistently improves the performance of both methods. For scientific reasoning (SciQ) and physical commonsense (PiQA) tasks, `LinUpper` strategy also enhances both baseline methods, with the most substantial improvement observed in SciQ (from 52.8% to 53.2% with DoGE and from 53.3% to 54.5% with DoReMi). These results indicate that `LinUpper` is particularly effective in tasks requiring nuanced reasoning and factual knowledge.

## 6.4 EXPERIMENTS ON 1.4B AND 7B LLAMA MODELS

We conduct experiments to train 1.4B (billion) and 7B parameter models with Llama architecture on randomly sampled subsets of the FineWeb 15T dataset. For the 1.4B and 7B parameter models, we use approximately 100B and 175B randomly sampled tokens from the FineWeb 15T dataset, respectively. We report the benchmark results on the last checkpoints.

Table 3: 5-shot performance evaluations under *non-uniform* domain sampling for the GPT2-medium model. Our `LinUpper` instance reweighting approach creates synergies with state-of-the-art domain reweighting techniques.

| Benchmark Accuracy | Method | | | |
|---|---|---|---|---|
| | DoGE | DoGE + LinUpper | DoReMi | DoReMi + LinUpper |
| LogiQA | 27.2 | **28.6** | 27.2 | **27.6** |
| LogiQA 2 | 27.5 | **28.0** | 27.7 | **27.9** |
| SciQ | 52.8 | **53.2** | 53.3 | **54.5** |
| PiQA | 55.8 | **56.3** | 55.7 | **56.1** |
| *Mean* | 40.8 | **41.5** | 41.0 | **41.5** |

Table 4: Average performance of the 1.4B parameter model across LUR and QA benchmarks. Overall, our `LinUpper` method improves results for 11 out of 19 tasks with an average performance gain of 0.66% among LUR tasks and 1.72% among QA tasks.

Table 5: Results of the 7B parameter model. Overall, our `LinUpper` method improves results for 14 out of 19 tasks with an average performance gain of 1.44% among LUR tasks and 1.16% among QA tasks.

| Avg. Performance | Uniform | LinUpper (ours) | Difference |
|---|---|---|---|
| LUR (13 tasks) | 48.51 | **49.16** | 0.66 |
| QA (6 tasks) | 42.35 | **44.07** | 1.72 |
| *Across 19 tasks* | *46.56* | ***47.55*** | *0.99* |

| Avg. Performance | Uniform | LinUpper (ours) | Difference |
|---|---|---|---|
| LUR (13 tasks) | 50.54 | **51.98** | 1.44 |
| QA (6 tasks) | 48.15 | **49.31** | 1.16 |
| *Across 19 tasks* | *49.79* | ***51.14*** | *1.35* |

For these larger models, we use the question-answering (QA) benchmarks and additionally employ language understanding and reasoning (LUR) tasks since their capacity allows for more complex tasks. Please see Appendix A for more details, including hyperparameters, model architectural details, and performance on the full list of benchmarks.

Tables 4 and 5 show the average performance among LUR and QA tasks for the 1.4B and 7B parameter models, respectively. For the 1.4B parameter model, we see that our `LinUpper` strategy improves the average model performance by 0.66% and 1.72% for LUR and QA tasks, respectively. In particular, we note that our `LinUpper` method improves performance for 11 out of 19 tasks in total (see Tables 8 and 9 in Appendix A.3). For the 7B parameter model, our `LinUpper` method improves results for 14 out of 19 tasks (see Tables 10 and 11 in Appendix A.3) with an average performance gain of 1.44% among LUR tasks and 1.16% among QA tasks. These improvements are particularly noteworthy given the scale of these models, highlighting the scalability and effectiveness of our method.

## 7 CONCLUSION

In this paper, we introduce a novel sample-level reweighting framework aimed at improving the efficiency and effectiveness of large language model (LLM) pretraining. By dynamically adjusting the importance of individual samples based on their loss values, our approach overcomes the limitations of traditional uniform averaging methods and adds a new dimension to existing domain-level reweighting methods by incorporating more fine-grained sample-level dynamics. Through extensive theoretical and empirical validation, we demonstrate that down-weighting low-loss samples accelerates convergence and improves performance. The experiments show that our proposed `LinUpper` strategy consistently outperforms uniform sampling on common LLM reasoning and commonsense benchmarks. Similarly, our approach creates synergies with existing domain reweighting techniques, which underpins the importance of sample-level dynamics. We observe that the benefits of our method are more pronounced in larger models, while smaller models show less significant improvements on benchmarks. This could be due to the fact that smaller models may have limited capacity to fully exploit the advantages of our approach. However, our overall findings highlight the potential of loss-based reweighting strategies to optimize LLM pretraining both in training efficiency and in model evaluation performance.

ACKNOWLEDGMENTS

We thank the IBM Data Engineering Team for their support in running experiments with 1.4B and 7B parameter models. The work of D. Sow and Y. Liang was supported in part by the U.S. National Science Foundation with the grants ECCS-2113860, ECCS-2413528, and CNS-2112471. The work of H. Woisetschläger was partially supported by the Bavarian State Ministry of Economic Affairs, Regional Development and Energy with grant DIK0446/01.

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

# Appendix

## A EXPERIMENTAL DETAILS

For full reproducibility of our work we provide insights on extending existing training routines with our minimal-invasive loss reweighting scheme.

### A.1 REFERENCE IMPLEMENTATION & REPRODUCIBILITY

To integrate our loss reweighting method with a PyTorch training routine, we have to add two plug-and-play components. First, we need to define four **utility functions** to ensure the proper functionality of our approach with reference implementations provided in Listing 1:

1. `apply_strategy(...)`: Implements different reweighting strategies such as focusing on medium losses (quadratic), capping high losses (linupper), or emphasizing extreme values (extremes).
2. `scale_losses(...)`: Applies exponential scaling to adjust the importance of losses based on a temperature parameter $r$.
3. `normalize_losses(...)`: Normalizes losses to a bounded interval $[-\delta, \delta]$ for consistency in subsequent reweighting.
4. `get_batch_loss_from_logits(...)`: Computes per-sample losses using cross-entropy, ensuring padding tokens do not affect the results.

These four functions are then employed inside the training step. We need to capture the unprocessed raw losses and apply our reweighting scheme. We provide a reference implementation for PyTorch FSDP training in Listing 2 to enable multi-GPU training.

For a full reference, we provide a ready-to-use implementation on GitHub for the GPT2 experiments: `https://github.com/sowmaster/Sample-Level-Loss-Reweighting-ICLR-2025`.

Listing 1: Utility functions

```python
import torch

# Exponential scaling function
def scale_losses(losses, r):
    return torch.exp(losses / r)

# Normalization function for losses
def normalize_losses(losses, delta=1., l_min=0., l_max=1.):
    return 2. * delta * losses / max(l_max - l_min, 1e-6) - delta * (
        l_max + l_min) / max(l_max - l_min, 1e-6)

# Reweighting strategies
def apply_strategy(losses, delta=1.0, strategy="linupper"):
    if strategy == "linupper":
        return torch.minimum(losses + delta, delta * torch.ones_like(
            losses))
    elif strategy == "uniform":
        # We do not reweight here. This is our baseline.
        return losses
    elif strategy == "quadratic":
        return 1 - losses**2 / delta**2
    elif strategy == "extremes":
        return torch.abs(losses)
    else:
        raise NotImplementedError

# Compute batch losses from logits
def get_batch_loss_from_logits(logits, labels):
    ignore_index = -100
    shift_logits = logits[..., :-1, :].contiguous()
    shift_labels = labels[..., 1:].contiguous()
```

```
30      num_active = (shift_labels != ignore_index).sum(dim=1)
31      loss_fct = torch.nn.CrossEntropyLoss(reduction='none')
32      loss = loss_fct(shift_logits.view(-1, logits.size(-1)), shift_labels.
            view(-1).long())
33      return loss.view(logits.size(0), -1).sum(dim=1) / num_active,
            num_active
```

Listing 2: Integration of reweighting strategies into a multi-GPU training loop.

```
1   import torch
2   import torch.distributed as dist
3
4   # Define reweighting function mapping
5   STRATEGY = "linupper"
6
7   # This has to be placed inside a training step.
8   ...
9
10  # Assume model outputs (logits) and labels are already computed
11  device_losses, len_norms = get_batch_loss_from_logits(outputs, labels)
12
13  # Initialize placeholder to store the losses from all GPUs
14  gathered_losses = torch.zeros(
15      dist.get_world_size(),
16      len(device_losses),
17      device=device_losses.device,
18      dtype=device_losses.dtype
19  )
20
21  # Gather losses across all GPUs into tenor
22  dist.all_gather_into_tensor(gathered_losses, device_losses.detach())
23
24  r = r_scheduler(step, cfg.initial_r)
25  # Compute sample weights
26  with torch.no_grad():
27      min_loss = gathered_losses.min().item()
28      max_loss = gathered_losses.max().item()
29      normalized_losses = normalize_losses(gathered_losses.view(-1), delta
            =1., l_min=min_loss, l_max=max_loss)
30      reweighted_losses = apply_strategy(normalized_losses, delta=1.,
            strategy=STRATEGY)
31      scaled_losses = scale_losses(reweighted_losses - reweighted_losses.
            max().item(), l=r)
32      weights = scaled_losses / scaled_losses.sum()
33
34      # for instance local_rank can be obtained with dist.get_rank()
35      device_weights = weights.view(dist.get_world_size(), -1)[local_rank,
            :]
36
37  # Reweight losses and scale appropriately
38  loss = torch.sum(device_weights * device_losses) * dist.get_world_size()
39  loss.backward()
```

## A.2 TRAINING & EVALUATION DETAILS

In the following, we outline the hyper-parameter configuration and hardware requirements to fully reproduce our experiments.

**Models**. We employ three different *GPT2 models* to study the effectiveness of our reweighting approach on various model sizes. We report the architectural details in Table 6. Furthermore, to demonstrate the performance of our reweighting approach on state-of-the-art models, we employ a 1.4B parameter model with *Llama* architecture.

Table 6: Model Architecture

| Architecture | Name | Parameters | Layers | Attention Heads | Embedding Dimensions | Hidden Dimensions | Max. Sequence Length |
|---|---|---|---|---|---|---|---|
| GPT-2 | Mini | 124M | 12 | 12 | 768 | 3072 | 512 |
| | Small | 210M | 24 | 16 | 786 | 3072 | 512 |
| | Medium | 300M | 36 | 24 | 786 | 3072 | 512 |
| LLAMA | 1.4B | 1400M | 24 | 16 | 2048 | 2048 | 8192 |
| | 7B | 7000M | 32 | 32 | 4096 | 4096 | 8192 |

**Datasets**. As a dataset for training our GPT2 models, we employ the SlimPajama-6B dataset, consisting of 7 domain partitions (ArXiv, Book, CC, C4, GitHub, StackExchange, and Wikipedia). The exact number of documents seen is reported in Table 7. We use the FineWeb 15T dataset to train the larger Llama-1.4B and Llama-7B models. For the 1.4B and 7B models, we use approximately 100B and 175B randomly sampled tokens from the FineWeb 15T dataset, respectively.

**Pretraining procedure**. We pretrain all models with masked language modeling and do not apply any instruction tuning routine for subsequent evaluations. We set $r$ to 100 for a specific number of warmup steps and then reduce it to 1 for the remaining steps. The GPT2 models are trained for 20,000 steps in total, and the Llama models are trained for 100,000 and 170,000 steps, respectively for the 1.4B and 7B model. More hyperparameters are provided in Table 7. For the GPT2 models, we use the Huggingface `trainer` interface and the `accelerate` library for distributed training. For the Llama models, we use fully sharded data parallel (FSDP).

Table 7: Training Hyperparameters for our benchmark evaluations

| Architecture | Name | Minibatch Size | Learning Rate (LR) | Weight Decay | LR Schedule | LR End | Warmup Steps | Max. Grad Norm | r | Training Steps | Total Documents Seen |
|---|---|---|---|---|---|---|---|---|---|---|---|
| GPT-2 | Mini | 32 | 0.0005 | 0.01 | Linear Warmup Cosine | 0.0001 | 500 | 1.0 | 0.4 | 20,000 | 640,000 |
| | Small | 48 | 0.0005 | 0.01 | Linear Warmup Cosine | 0.0001 | 500 | 1.0 | 0.4 | 20,000 | 960,000 |
| | Medium | 48 | 0.0005 | 0.01 | Linear Warmup Cosine | 0.0001 | 500 | 1.0 | 0.4 | 20,000 | 960,000 |
| LLAMA | 1.4B | 128 | 0.0003 | 0.01 | Linear Warmup Cosine | 0.00003 | 2,000 | 1.0 | 1.0 | 100,000 | 12,800,000 |
| | 7B | 128 | 0.0003 | 0.01 | Linear Warmup Cosine | 0.00003 | 2,000 | 1.0 | 2.0 | 170,000 | 21,760,000 |

**Benchmarks**. We use the LM Eval Harness to generate our benchmark results. We evaluate the GPT2 models every 2000 steps and the Llama model every 5000 steps. For the GPT2 models, we find that only question-answering (QA) tasks work well at this scale, which is why we report 5-shot results on LogiQA, LogiQA-2, SciQ, and PiQA. All other benchmarks yielded random results at this scale, i.e., they were not useful for model evaluation. The reason is the limited capacity of the GPT2 models compared to larger models like the Llama-1.4B or 7B models. For the Llama architecture models, we employ 13 language understanding and reasoning (LUR) and 6 QA tasks to provide a holistic picture.

## A.3  ADDITIONAL EXPERIMENTAL RESULTS

**Benchmark results**. Using our `LinUpper` method, the experiments with the 1.4B parameter Llama model show an average performance gain of $0.66\%$, improving on 6 out of 13 benchmarks and tied on one benchmark (Table 8). For QA benchmarks, we improved on 5/6 tasks with an average of $1.72\%$ (Table 9).

We observe that our **LinUpper** strategy also improves the performance of the 7B model. Our strategy outperforms the baseline in 9/13 LUR (Table 10) and 5/6 QA tasks (Table 11), respectively. On average, across all tasks, our **LinUpper** strategy improves performance by $1.44\%$ for LUR tasks and $1.16\%$ for QA tasks.

**Perplexity**. We present additional perplexity visualizations of our GPT2 pretraining runs. Figure 3 shows the results for GPT2-mini and Figure 4 shows the results for GPT2-small.

Table 8: Results of 1.4B parameter model on language understanding and reasoning benchmarks. Overall, our `LinUpper` method improves the results for 6 out of 13 tasks, and there is a tie for one task, with an average performance gain of 0.66%.

| Benchmark Name | Uniform | LinUpper (ours) | Difference |
|---|---|---|---|
| ARC Challenge | 33.19 | 33.02 | -0.17 |
| ARC Easy | 63.80 | 63.38 | -0.42 |
| COPA | 77.0 | 77.0 | 0.00 |
| Lambada (OpenAI) | 49.87 | 49.60 | -0.27 |
| Lambada (Standard) | 45.20 | 43.30 | -1.90 |
| MMLU | 26.04 | 25.49 | -0.54 |
| MNLI | 31.91 | 36.92 | 5.01 |
| MNLI (mismatch) | 32.07 | 36.56 | 4.50 |
| RTE | 51.26 | 52.35 | 1.08 |
| SST-2 | 66.86 | 59.40 | -7.45 |
| TinyARC | 40.46 | 40.85 | 0.39 |
| TinyWinoGrande | 56.17 | 62.11 | 5.93 |
| WinoGrande | 56.75 | 59.12 | 2.37 |
| *Mean* | *48.51* | *49.16* | *0.66* |

Table 9: Results of 1.4B parameter model on language question answering benchmarks. Overall, our `LinUpper` method improves results for 5 out of 6 tasks with an average performance gain of 1.72%.

| Benchmark Name | Uniform | LinUpper (ours) | Difference |
|---|---|---|---|
| BoolQ | 61.59 | 65.47 | 3.88 |
| LogiQA | 20.28 | 23.66 | 3.38 |
| LogiQA2 | 26.02 | 27.48 | 1.46 |
| SciQ | 91.30 | 91.90 | 0.60 |
| SocialIQA | 43.86 | 45.19 | 1.33 |
| TriviaQA (Exact Match) | 11.05 | 10.73 | -0.32 |
| *Mean* | *42.35* | *44.07* | *1.72* |

Table 10: Results of the 7B parameter model on language understanding and reasoning benchmark tasks. Overall, our `LinUpper` method improves results for 9 out of 13 tasks with an average performance gain of 1.44%.

| Benchmark Name | Uniform | LinUpper (ours) | Difference |
|---|---|---|---|
| ARC Challenge | 41.81 | 39.42 | -2.39 |
| ARC Easy | 72.77 | 74.24 | 1.47 |
| COPA | 83.00 | 84.00 | 1.00 |
| Lambada (OpenAI) | 61.28 | 62.43 | 1.14 |
| Lambada (Standard) | 58.08 | 58.80 | 0.72 |
| MMLU | 24.33 | 25.81 | 1.47 |
| MNLI | 32.28 | 34.24 | 1.97 |
| MNLI (mismatch) | 32.35 | 34.01 | 1.66 |
| RTE | 50.54 | 49.10 | -1.44 |
| SST-2 | 49.89 | 69.38 | 19.50 |
| TinyARC | 36.17 | 32.80 | -3.37 |
| TinyWinoGrande | 49.09 | 44.20 | -4.88 |
| WinoGrande | 65.43 | 67.32 | 1.89 |
| *Mean* | *50.54* | *51.98* | *1.44* |

Table 11: Results of the 7B parameter model on question answering benchmark tasks. Overall, our `LinUpper` method improves results for 5 out of 6 tasks with an average performance gain of 1.16%.

| Benchmark Name | Uniform | LinUpper (ours) | Difference |
|---|---|---|---|
| BoolQ | 73.52 | 74.40 | 0.89 |
| LogiQA | 23.35 | 27.65 | 4.30 |
| LogiQA2 | 26.72 | 27.86 | 1.15 |
| SciQ | 95.20 | 95.00 | -0.20 |
| SocialIQA | 47.80 | 48.11 | 0.31 |
| TriviaQA (Exact Match) | 22.30 | 22.79 | 0.49 |
| *Mean* | *48.15* | *49.31* | *1.16* |

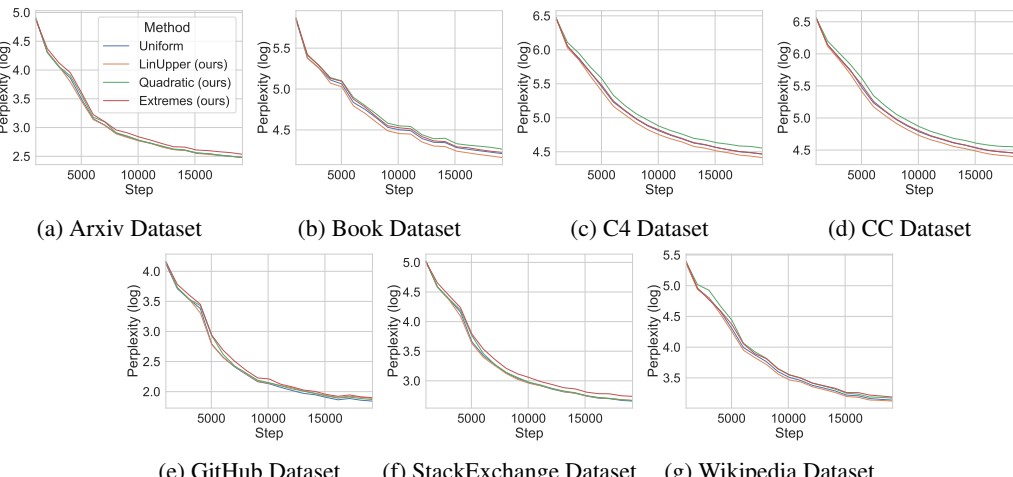

(a) Arxiv Dataset    (b) Book Dataset    (c) C4 Dataset    (d) CC Dataset

(e) GitHub Dataset    (f) StackExchange Dataset    (g) Wikipedia Dataset

Figure 3: Per-domain perplexities on hold-out validation sets under the uniform domain sampling setting for the GPT2-mini model. Our reweighting strategy `LinUpper` strategy achieves better or at least comparable perplexity on 6 out of 7 domains.

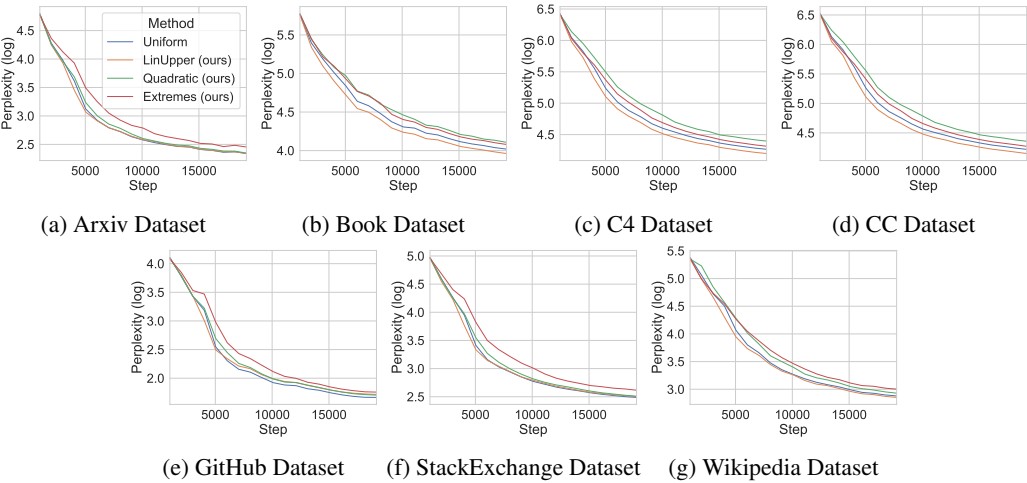

Figure 4: Per-domain perplexities on hold-out validation sets under the uniform domain sampling setting for the GPT2-small model. Our reweighting strategy `LinUpper` strategy achieves better or at least comparable perplexity on 6 out of 7 domains.

### A.4 SENSITIVITY TO THE VALUE OF $r$

We conduct experiments using different values of $r$ to understand the sensitivity of our methods to the value of $r$. The perplexity plots for out method `LinUpper` with different values of $r$ are provided in Figure 5, which shows that when the value of $r$ is large (e.g. $r = 1$) then, as expected, the performance of our method becomes closer to that of the uniform baseline. Decreasing the value of $r$ leads to the diminished effect of low-loss samples, but this can also have a negative effect on the performance when $r$ is too small (e.g. $r = 0.2$), potentially due to data wastage by over filtering the low-loss samples.

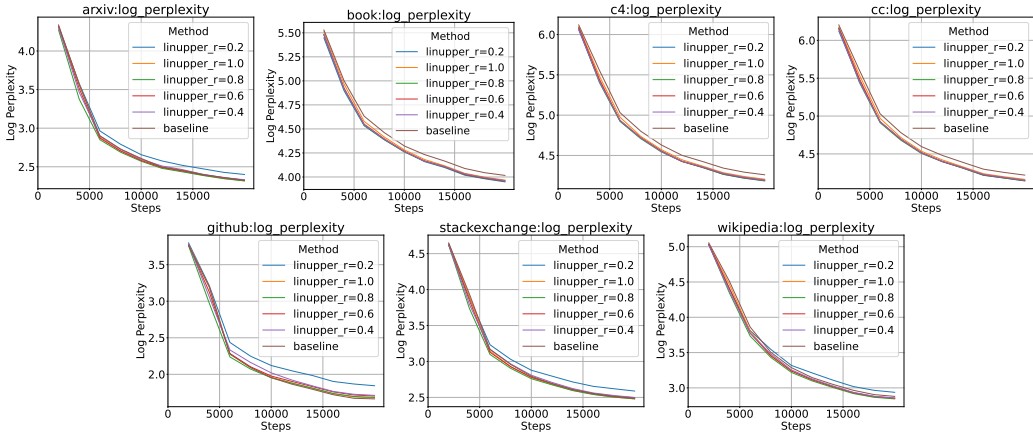

Figure 5: Sensitivity of our method LinUpper to the value of $r$. When the $r$ is large (e.g. $r = 1$) the performance of our method becomes closer to that of the uniform baseline. Decreasing the value of $r$ leads to diminished effect of low-loss samples, but can also have a negative effect on the performance when $r$ is too small (eg. $r = 0.2$)

### A.5 FURTHER DISCUSSIONS ABOUT OUR REWEIGHTING UPPER BOUND

To demonstrate that in practice the upper bound of $w_i < \frac{2}{b}$ is, in fact, satisfied by our LinUpper method, we provide the distribution of the maximum weights per step (see Figure 6) during the pretraining of the 7B parameter model with our LinUpper method. In this experiment, the minibatch size is 128 and hence the upper bound is $\frac{2}{128} = 0.0156$. As depicted in Figure 6, the maximum

weight per step (which is around $0.01$) by our LinUpper method is clearly below the upper bound of $0.0156$.

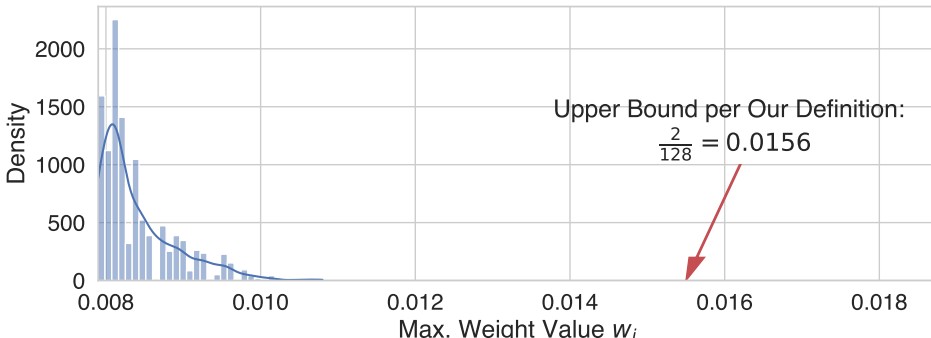

Figure 6: Distribution of maximum weight values when training our 7B parameter model with Llama architecture. The maximum weight values always stay well below the theoretical upper bound defined in Theorem 2.

# B    ADDITIONAL DISCUSSIONS

## B.1    TOY REGRESSION PROBLEM SETUP

To further illustrate the effectiveness of our reweighting strategies on simple ML problems, we apply them to a regression problem where the goal is to learn a linear mapping from input features to output labels, while dealing with noisy data and outliers. We generate a synthetic dataset as follows.

The clean input features $\mathbf{X}_{\text{clean}} \in \mathbb{R}^{n \times p}$ are sampled from the standard normal distribution, where $p$ is the dimensionality of the data, $n$ is the number of clean data points. The target output $\mathbf{y} \in \mathbb{R}^n$ for the clean data is generated using the following ground truth linear model with added noise:

$$\mathbf{y}_{\text{clean}} = \mathbf{X}_{\text{clean}} \mathbf{W}^* + b^* \mathbf{1}_n + c \cdot \boldsymbol{\epsilon},$$

where $\mathbf{W}^* \in \mathbb{R}^p$ and $b^* \in \mathbb{R}$ are the true model parameters, $\boldsymbol{\epsilon} \sim \mathcal{N}(\mathbf{0}_n, \mathbf{I}_n)$ represents Gaussian noise, and $c$ is a small constant controlling the noise level. For the $m$ outlier data, we generate each input features of $\mathbf{X}_{\text{ood}} \in \mathbb{R}^{m \times p}$ as $0.1 \cdot \mathcal{N}(0, 1) + 2.0$ with the corresponding target outputs $\mathbf{y}_{\text{ood}}$ drawn randomly from a standard normal distribution, i.e., $\mathbf{y}_{\text{ood}} \sim \mathcal{N}(0, 1)$.

The full training set consists of both the clean data $(\mathbf{X}_{\text{clean}}, \mathbf{y}_{\text{clean}})$ and the outlier data $(\mathbf{X}_{\text{ood}}, \mathbf{y}_{\text{ood}})$, concatenated together:

$$\mathbf{X}_{\text{all}} = [\mathbf{X}_{\text{clean}}; \mathbf{X}_{\text{ood}}], \quad \mathbf{y}_{\text{all}} = [\mathbf{y}_{\text{clean}}; \mathbf{y}_{\text{ood}}].$$

By incorporating outliers into the training set, we test the robustness of our reweighting strategies, which should down-weight the influence of noisy or irrelevant samples, allowing the model to learn the underlying clean data distribution more effectively.

We define the objective function for this regression task as the mean squared error between the model predictions and the target outputs. Given model parameters $\mathbf{W} \in \mathbb{R}^p$ and $b \in \mathbb{R}$, the loss for a single data point $(\mathbf{x}_i, y_i)$ computed as:

$$\ell_i(\mathbf{W}, b) = \frac{1}{2} \left( \mathbf{x}_i^\top \mathbf{W} + b - y_i \right)^2. \tag{11}$$

To evaluate model performance, we compute the mean squared error on a held-out test set $(\mathbf{X}_{\text{test}}, \mathbf{y}_{\text{test}})$, which was generated using the same process as the clean training data. We use $p = 64$, $n = 3200$, and $m = 800$.

## B.2    COMPARISON BETWEEN DIFFERENT REWEIGHTING METHODS

The results from Figure 7 clearly show that `LinUpper` outperforms all other reweighting strategies and the uniform baseline, achieving the fastest convergence. This supports our theoretical findings that down-weighting low-loss samples allows the model to focus on more informative examples, thus accelerating the optimization. Interestingly, the `Quadratic` strategy also performs better than the uniform baseline, indicating that selectively up-weighting the moderate loss samples can improve performance. While `Quadratic` converges more slowly than `LinUpper`, it still highlights that more balanced reweighting can provide benefits over uniform sampling when dealing with noisy data.

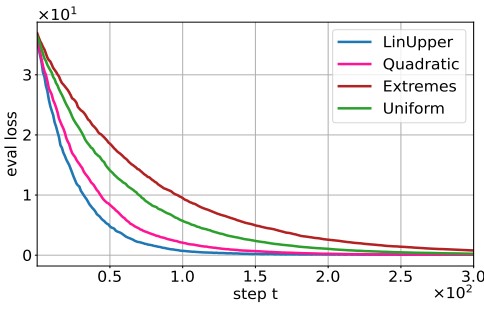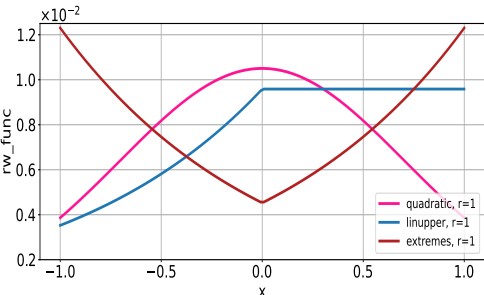

Figure 7: **Left:** Test loss vs training steps for the toy regression problem. Our `LinUpper` strategy achieves faster convergence compared to all the other methods. Notably, the `Quadratic` strategy also outperforms the uniform baseline, further highlighting the benefit of selective reweighting. **Right:** Shape of the different reweighting methods used with $r = 1$.

### B.3 COMPARISON WITH ROBUST OPTIMIZATION METHODS

Comparing with traditional robust optimization methods, such as those based on distributionally robust optimization (DRO), our methods impose a cap on sample weights ($w_i \leq 2/M$), as justified by our theoretical analysis (Proposition 1). This cap prevents over-reliance on a small subset of high-loss samples, which can lead to overfitting to outliers. In contrast, DRO methods often focus heavily on worst-case scenarios, potentially leading to suboptimal generalization when the training data contains a substantial amount of noisy samples. To demonstrate this benefit of our method in practice, we conducted experiments to compare with the traditional KL-divergence regularized distributionally robust optimization (DRO-KL).

First, we created a synthetic regression problem (similar to the regression problem described above) with 25% of the data being outliers to compare the robustness of the different methods against outliers. As shown in Figure 8, we note the following: because it focuses heavily on the hard samples, `DRO-KL` diverges when we use the same learning rate as the other two methods (our `LinUpper` method & the `Uniform` baseline). The `DRO-KL` method requires a smaller learning rate to converge, in which case it converges slower than our method. Furthermore, we include the `DRO-KL` method among the compared baselines for the GPT2-medium experiments, as shown in Table 12. On average, our `LinUpper` strategy outperforms the other `DRO-KL` configurations, which we attribute to its ability to handle outliers.

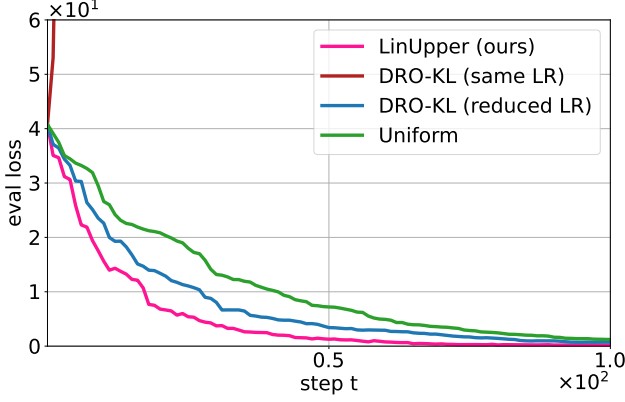

Figure 8: Regression problem with `DRO-KL`.

Table 12: Comparison of our `LinUpper` loss reweighting strategy with various `DRO-KL` configurations on the GPT2-medium experimental setup. On average, our `LinUpper` strategy outperforms the other `DRO-KL` configurations.

| | Uniform | LinUpper (r = 0.4; ours) | DRO (KL = 0.4) | DRO (KL = 1.0) | DRO (KL = 2.0) | DRO (KL = 5.0) | DRO (KL = 10.0) |
|---|---|---|---|---|---|---|---|
| LogiQA | 25.7 | **27.9** | 23.2 | 25.5 | 26.8 | 25.9 | 26.3 |
| LogiQA2 | 27.5 | 27.6 | 24.4 | **28.7** | 27.5 | 28.2 | 27.6 |
| SciQ | 49.0 | **51.8** | 41.1 | 48.6 | 49.4 | 50.5 | 49.4 |
| PiQA | 55.5 | 56.2 | 54.0 | 56.2 | 55.7 | **56.5** | 55.4 |
| *Mean* | *39.4* | ***40.9*** | *35.7* | *39.8* | *39.9* | *40.3* | *39.7* |

## C PROOFS

**Lemma 1.** *Let Assumptions 1 and 2 hold. Then we have*

$$\left\| \nabla f(\theta) - \nabla f(\theta') \right\|^2 \le 2L \left( f(\theta) - f(\theta') - \langle \nabla f(\theta'), \theta - \theta' \rangle \right).$$

*Proof.* Consider the following

$$f(\theta_1) - f(\theta_2) = f(\theta_1) - f(\theta) + f(\theta) - f(\theta_2). \tag{12}$$

From the $L$-smoothness of $f$, we have for all $\theta_2, \theta \in \mathbb{R}^d$,

$$f(\theta) - f(\theta_2) \le \langle \nabla f(\theta_2), \theta - \theta_2 \rangle + \frac{L}{2} \left\| \theta - \theta_2 \right\|^2. \tag{13}$$

From the the convexity of $f$, we have for all $\theta_1, \theta \in \mathbb{R}^d$,

$$f(\theta_1) - f(\theta) \le \langle \nabla f(\theta_1), \theta_1 - \theta \rangle. \tag{14}$$

Substituting Eq. (13), Eq. (14) in Eq. (12), we get

$$f(\theta_1) - f(\theta_2) \le \langle \nabla f(\theta_1), \theta_1 - \theta \rangle + \langle \nabla f(\theta_2), \theta - \theta_2 \rangle + \frac{L}{2} \left\| \theta - \theta_2 \right\|^2. \tag{15}$$

Let $\theta = \theta_2 - \frac{1}{L} \left( \nabla f(\theta_2) - \nabla f(\theta_1) \right)$. Substituting the definition of $\theta$ in Eq. (15), we obtain

$$
\begin{aligned}
f(\theta_1) - f(\theta_2) &\le \langle \nabla f(\theta_1), \theta_1 - \theta_2 + \frac{1}{L} \left( \nabla f(\theta_2) - \nabla f(\theta_1) \right) \rangle - \frac{1}{L} \langle \nabla f(\theta_2), \nabla f(\theta_2) - \nabla f(\theta_1) \rangle \\
&\quad + \frac{1}{2L} \left\| \nabla f(\theta_2) - \nabla f(\theta_1) \right\|^2 \\
&= \langle \nabla f(\theta_1), \theta_1 - \theta_2 \rangle - \frac{1}{L} \| \nabla f(\theta_2) - \nabla f(\theta_1) \|^2 + \frac{1}{2L} \left\| \nabla f(\theta_2) - \nabla f(\theta_1) \right\|^2 \\
&= \langle \nabla f(\theta_1), \theta_1 - \theta_2 \rangle - \frac{1}{2L} \| \nabla f(\theta_2) - \nabla f(\theta_1) \|^2. 
\end{aligned} \tag{16}
$$

$\square$

From lemma 1, we obtain at the optimum $\theta^*$

$$\left\| \nabla f(\theta) - \nabla f(\theta^*) \right\|^2 \le 2L \left( f(\theta) - f(\theta^*) \right).$$

### C.1 FULL GRADIENT

We have

$$
\begin{aligned}
\left\| \theta^{t+1} - \theta^* \right\|^2 &= \left\| \theta^t - \theta^* \right\|^2 + \left\| \theta^{t+1} - \theta^t \right\|^2 + 2 \left\langle \theta^{t+1} - \theta^t, \theta^t - \theta^* \right\rangle \\
&= \left\| \theta^t - \theta^* \right\|^2 + \left\| \eta_t \sum_{i=1}^{M} w_{i,t} \nabla f_i(\theta^t) \right\|^2 - 2 \left\langle \eta_t \sum_{i=1}^{M} w_{i,t} \nabla f_i(\theta^t), \theta^t - \theta^* \right\rangle \\
&\le \left\| \theta^t - \theta^* \right\|^2 + \eta_t^2 M \sum_{i=1}^{M} w_{i,t}^2 \left\| \nabla f_i(\theta^t) \right\|^2 - 2\eta_t \sum_{i=1}^{M} w_{i,t} \left\langle \nabla f_i(\theta^t), \theta^t - \theta^* \right\rangle
\end{aligned} \tag{17}
$$

Using the convexity of function $f_i$, we have

$$\left\langle \nabla f_i(\theta^t), \theta^t - \theta^* \right\rangle \ge f_i(\theta^t) - f_i(\theta^*). \tag{18}$$

Therefore combining, we obtain

$$\left\| \theta^{t+1} - \theta^* \right\|^2 \le \left\| \theta^t - \theta^* \right\|^2 + \eta_t^2 M \sum_{i=1}^{M} w_{i,t}^2 \left\| \nabla f_i(\theta^t) \right\|^2 - 2\eta_t \sum_{i=1}^{M} w_{i,t} \left( f_i(\theta^t) - f_i(\theta^*) \right) \tag{19}$$

Next, we upper-bound the quantity $\left\|\nabla f_i(\theta^t)\right\|^2$.

$$\left\|\nabla f_i(\theta^t)\right\|^2 \leq 2\left\|\nabla f_i(\theta^t) - \nabla f_i(\theta^*)\right\|^2 + 2\left\|\nabla f_i(\theta^*)\right\|^2$$
$$\leq 4L\left(f_i(\theta^t) - f_i(\theta^*)\right) + 2\sigma_*^2 \tag{20}$$

where Eq. (20) follows from lemma 1.

Now, combining Eq. (19) and Eq. (20), we obtain

$$\left\|\theta^{t+1} - \theta^*\right\|^2 \leq \left\|\theta^t - \theta^*\right\|^2 + 4ML\eta_t^2 \sum_{i=1}^{M} w_{i,t}^2 \left(f_i(\theta^t) - f_i(\theta^*)\right) + 2M\eta_t^2\sigma_*^2 \sum_{i=1}^{M} w_{i,t}^2 \tag{21}$$

$$- 2\eta_t \sum_{i=1}^{M} w_{i,t} \left(f_i(\theta^t) - f_i(\theta^*)\right)$$

$$= \left\|\theta^t - \theta^*\right\|^2 + \sum_{i=1}^{M} \left(4ML\eta_t w_{i,t} - 2\right)\eta_t w_{i,t} \left(f_i(\theta^t) - f_i(\theta^*)\right)$$

$$+ 2M\eta_t^2\sigma_*^2 \sum_{i=1}^{M} w_{i,t}^2, \tag{22}$$

Selecting $\eta_t$ such that $4ML\eta_t w_{i,t} - 2 \leq -1 \ \forall \ i$, i.e., $\eta_t \leq \frac{1}{4MLw_{\max,t}}$, yields

$$\left\|\theta^{t+1} - \theta^*\right\|^2 \leq \left\|\theta^t - \theta^*\right\|^2 - \eta_t \sum_{i=1}^{M} w_{i,t} \left(f_i(\theta^t) - f_i(\theta^*)\right) + 2M\eta_t^2\sigma_*^2, \tag{23}$$

where we also used the fact that $\sum_{i=1}^{M} w_{i,t}^2 \leq 1$ due to $\sum_{i=1}^{M} w_{i,t} = 1$ and $w_{i,t} \geq 0$. Hence,

$$\left\|\theta^{t+1} - \theta^*\right\|^2$$

$$\leq \left\|\theta^t - \theta^*\right\|^2 - \frac{\eta_t}{M} \sum_{i=1}^{M} \left(f_i(\theta^t) - f_i(\theta^*)\right) + \eta_t \sum_{i=1}^{M} \left(\frac{1}{M} - w_{i,t}\right) \left(f_i(\theta^t) - f_i(\theta^*)\right) + 2M\eta_t^2\sigma_*^2$$

$$= \left\|\theta^t - \theta^*\right\|^2 - \eta_t \left(f(\theta^t) - f(\theta^*)\right) + \eta_t \sum_{i=1}^{M} \left(\frac{1}{M} - w_{i,t}\right) \left(f_i(\theta^t) - f_i(\theta^*)\right) + 2M\eta_t^2\sigma_*^2. \tag{24}$$

Telescoping Eq. (24) from $t = 0$ to $T - 1$ yields

$$\left\|\theta^T - \theta^*\right\|^2 \leq \left\|\theta^0 - \theta^*\right\|^2 - \sum_{t=0}^{T-1} \eta_t \left(f(\theta^t) - f(\theta^*)\right) + \sum_{t=0}^{T-1} \eta_t \sum_{i=1}^{M} \left(\frac{1}{M} - w_{i,t}\right) \left(f_i(\theta^t) - f_i(\theta^*)\right)$$

$$+ 2M\sigma_*^2 \sum_{t=0}^{T-1} \eta_t^2. \tag{25}$$

For a weighting scheme such that $w_{\max,t} \leq \frac{2}{M}$ so that $\eta_t$ can be fixed to $\eta = \frac{1}{8L}$, we obtain

$$\eta \sum_{t=0}^{T-1} \left(f(\theta^t) - f(\theta^*)\right) \leq \left\|\theta^0 - \theta^*\right\|^2 + \eta \sum_{t=0}^{T-1} \sum_{i=1}^{M} \left(\frac{1}{M} - w_{i,t}\right) \left(f_i(\theta^t) - f_i(\theta^*)\right) + 2M\sigma_*^2\eta^2 T.$$

$$f(\bar{\theta}^T) - f(\theta^*) \leq \frac{8L\left\|\theta^0 - \theta^*\right\|^2}{T} + \frac{1}{T} \sum_{t=0}^{T-1} \sum_{i=1}^{M} \left(\frac{1}{M} - w_{i,t}\right) \left(f_i(\theta^t) - f_i(\theta^*)\right) + \frac{M\sigma_*^2}{4L},$$

where $\bar{\theta}^T = \frac{1}{T} \sum_{t=0}^{T-1} \theta^t$. Define $\delta^t = \sum_{i=1}^{M} \left(\frac{1}{M} - w_{i,t}\right) \left(f_i(\theta^t) - f_i(\theta^*)\right)$. Hence, we obtain

$$f(\bar{\theta}^T) - f(\theta^*) \leq \frac{8L\left\|\theta^0 - \theta^*\right\|^2}{T} + \frac{1}{T} \sum_{t=0}^{T-1} \delta^t + \frac{M\sigma_*^2}{4L}. \tag{26}$$

## C.2 OPTIMAL WEIGHT STRATEGY

**Proposition 1.** *The optimal strategy that minimizes the KL-divergence regularized $\delta^t$, i.e., $\delta^t + r\sum_{i=1}^{M} w_i \log(Mw_i)$, is given by*

$$w_i = C \min\left\{ \exp\left(\frac{h_i}{r}\right), \frac{2}{M} \right\}, \tag{5}$$

*where $C$ is a normalizing constant that ensures $\sum_i w_i = 1$.*

*Proof.* For the sake of clarity, we drop the index $t$ in $\delta^t$ in the following proof.

Let $\Delta_i = f(\mathbf{x}_i; \theta^t) - f(\mathbf{x}_i; \theta^*)$ be the loss gap. After dropping all terms that do not depends on the weights $w_i$'s, we have the following constrained optimization problem

$$\min_{w_1,\ldots,w_M} \quad -\sum_{i=1}^{M} w_i \Delta_i + r \sum_{i=1}^{M} w_i \log w_i \tag{27}$$

$$\text{subject to} \quad w_i \leq \frac{2}{M}, i \in [M], \tag{28}$$

$$\sum_{i=1}^{M} w_i = 1. \tag{29}$$

As we have inequality constraints in the above problem, we introduce the slack variable $t$ and consider the Lagrangian function to solve the above problem as follows:

$$L(w_1,\ldots,w_M,\nu,\mu) = -\sum_{i=1}^{M} w_i \Delta_i + r \sum_{i=1}^{M} w_i \log w_i - \nu\left(\sum_{i=1}^{M} w_i - 1\right) - \mu\left(w_i - \frac{2}{M} + t^2\right), \tag{30}$$

and we have the gradients

$$\frac{\partial L}{\partial w_i} = -\Delta_i + r + r \log w_i - \nu - \mu$$

$$\frac{\partial L}{\partial \lambda} = -\left(\sum_{i=1}^{M} w_i - 1\right)$$

$$\frac{\partial L}{\partial \mu} = -\left(w_i - \frac{2}{M} + t^2\right). \tag{31}$$

We consider the complementary slackness condition. We have one slack variable $t$, and the corresponding Lagrange multiplier is $\nu$. We now consider whether a slack variable is zero (which the corresponding inequality constraint is active) or the Lagrange multiplier is zero (the constraint is inactive), we obtain

- $\mu = 0, t^2 > 0$: Using $\frac{\partial L}{\partial \nu} = 0$, we get $\sum_{i=1}^{M} w_i = 1$. Using $\frac{\partial L}{\partial w_i} = 0, \mu = 0$, we get $-\Delta_i + r + r \log w_i - \nu = 0$. Hence, we have $w_i = \exp\left(\frac{\nu + \Delta_i - r}{r}\right)$. Therefore, we obtain $\nu = -r \ln\left(\sum_{i=1}^{M} \exp\left(\frac{\Delta_i - r}{r}\right)\right)$ yielding $w_i = \exp\left(\frac{-r \ln\left(\sum_{i=1}^{M} \exp\left(\frac{\Delta_i - r}{r}\right)\right) + \Delta_i - r}{r}\right) = C \exp\left(\frac{\Delta_i}{r}\right)$ where $C = \exp\left(-\ln\left(\sum_{i=1}^{M} \exp\left(\frac{\Delta_i - r}{r}\right)\right) - 1\right)$.

- $\mu \neq 0, t^2 = 0$: Using $\frac{\partial L}{\partial \mu} = 0, t = 0$, we get $w_i = \frac{2}{M}$ and $\mu + \nu = -\Delta_i + r + r \log \frac{2}{M}$.

Hence, we obtain $w_i = \min\left\{C \exp\left(\frac{\Delta_i}{r}\right), \frac{2}{M}\right\}$. $\qquad\square$

## C.3 CONVEX MINIBATCH SGD

**Theorem 3** (Re-statement of Theorem 2). *Let Assumptions 1, 2, and 3 hold. Consider a minibatch of size $|\mathcal{B}| = b$ with reweighting scheme satisfying $\max_{i \in \mathcal{B}} w_{i,t} \leq 2/b$. Then,*

- **minibatch SGD:** *for $\eta = \frac{1}{8L\sqrt{T}}$, we have*

$$\mathbb{E}[f(\bar{\theta}^T) - f(\theta^*)] \leq \frac{8L\|\theta^0 - \theta^*\|^2}{\sqrt{T}} + \frac{1}{T} \sum_{t=0}^{T-1} \delta_t, \tag{32}$$

- **minibatch SGD with momentum:** *for $\eta = \frac{1}{8L\sqrt{T+1}}, \lambda_t > 0$, we have*

$$\mathbb{E}[f(\theta^T) - f(\theta^*)] \leq \frac{8L\|\theta^0 - \theta^*\|^2}{\sqrt{T+1}} + \frac{2}{T+1} \sum_{t=0}^{T-1} \delta_t + \frac{2}{T+1} \sum_{t=0}^{T-1} \lambda_t \mu_t, \tag{33}$$

*where $\delta_t = \mathbb{E}\left[\sum_{i \in \mathcal{B}} \left(\frac{1}{b} - w_{i,t}\right)(f_i(\theta^t) - f_i(\theta^*))\right]$, $\mu_t = \mathbb{E}\left[\sum_{i \in \mathcal{B}} \left(\frac{1}{b} - w_{i,t}\right)(f_i(\theta^t) - f_i(\theta^{t-1}))\right]$, and $\bar{\theta}^T = \frac{1}{T} \sum_{t=0}^{T-1} \theta^t$.*

*Proof.* (i) **Minibatch SGD:** For the minibatch SGD based on Eq. (2), we have

$$\|\theta^{t+1} - \theta^*\|^2 = \|\theta^t - \theta^*\|^2 + \|\theta^{t+1} - \theta^t\|^2 + 2\langle \theta^{t+1} - \theta^t, \theta^t - \theta^* \rangle$$

$$= \|\theta^t - \theta^*\|^2 + \left\|\eta_t \sum_{i \in \mathcal{B}} w_{i,t} \nabla f_i(\theta^t)\right\|^2 - 2\left\langle \eta_t \sum_{i \in \mathcal{B}} w_{i,t} \nabla f_i(\theta^t), \theta^t - \theta^* \right\rangle. \tag{34}$$

Applying expectation on both sides, we get

$$\mathbb{E}\|\theta^{t+1} - \theta^*\|^2$$

$$\leq \|\theta^t - \theta^*\|^2 + \eta_t^2 \mathbb{E}\left\|\sum_{i \in \mathcal{B}} w_{i,t} \nabla f_i(\theta^t)\right\|^2 - 2\eta_t \mathbb{E}\left\langle \sum_{i \in \mathcal{B}} w_{i,t} \nabla f_i(\theta^t), \theta^t - \theta^* \right\rangle$$

$$\overset{(a)}{\leq} \|\theta^t - \theta^*\|^2 + \eta_t^2 \mathbb{E}\left\|\sum_{i \in \mathcal{B}} w_{i,t} \nabla f_i(\theta^t)\right\|^2 - 2\eta_t \mathbb{E}\sum_{i \in \mathcal{B}} w_{i,t} \left(f_i(\theta^t) - f_i(\theta^*)\right)$$

$$\leq \|\theta^t - \theta^*\|^2 + \eta_t^2 b \mathbb{E}\left[\sum_{i \in \mathcal{B}} w_{i,t}^2 \|\nabla f_i(\theta^t)\|^2\right] - 2\eta_t \mathbb{E}\sum_{i \in \mathcal{B}} w_{i,t} \left(f_i(\theta^t) - f_i(\theta^*)\right)$$

$$\overset{(b)}{\leq} \|\theta^t - \theta^*\|^2 + 4L\eta_t^2 b \mathbb{E}\left[\sum_{i \in \mathcal{B}} w_{i,t}^2 (f_i(\theta^t) - f_i(\theta^*))\right] - 2\eta_t \mathbb{E}\sum_{i \in \mathcal{B}} w_{i,t} \left(f_i(\theta^t) - f_i(\theta^*)\right)$$

$$\leq \|\theta^t - \theta^*\|^2 + \mathbb{E}\left[\sum_{i \in \mathcal{B}} w_{i,t}(4L\eta_t^2 b w_{i,t} - 2\eta_t)(f_i(\theta^t) - f_i(\theta^*))\right], \tag{35}$$

where (a) follows from Assumption 1, and (b) follows from lemma 1. Selecting $\eta_t$ such that $4bL\eta_t w_{i,t} - 2 \leq -1 \ \forall i$, i.e., $\eta_t \leq \frac{1}{4bLw_{\max,t}}$, yields

$$\mathbb{E}\|\theta^{t+1} - \theta^*\|^2 \leq \|\theta^t - \theta^*\|^2 - \eta_t \mathbb{E}\left[\sum_{i \in \mathcal{B}} w_{i,t}(f_i(\theta^t) - f_i(\theta^*))\right]$$

$$= \|\theta^t - \theta^*\|^2 - \eta_t \mathbb{E}\left[\sum_{i \in \mathcal{B}} \frac{1}{b}(f_i(\theta^t) - f_i(\theta^*))\right]$$

$$+ \eta_t \mathbb{E}\left[\sum_{i \in \mathcal{B}} \left(\frac{1}{b} - w_{i,t}\right)(f_i(\theta^t) - f_i(\theta^*))\right]$$

$$= \left\|\theta^t - \theta^*\right\|^2 - \eta_t(f(\theta^t) - f(\theta^*)) + \eta_t \delta_t, \tag{36}$$

where $\delta_t = \mathbb{E}\left[\sum_{i \in \mathcal{B}} \left(\frac{1}{b} - w_{i,t}\right)(f_i(\theta^t) - f_i(\theta^*))\right]$ and $\sum_{i \in S} w_{i,t}^2 \leq 1$. Hence, telescoping Eq. (36) from $t = 0$ to $T - 1$ yields

$$\mathbb{E}\left\|\theta^T - \theta^*\right\|^2 \leq \left\|\theta^0 - \theta^*\right\|^2 - \sum_{t=0}^{T-1} \eta_t \left(f(\theta^t) - f(\theta^*)\right) + \sum_{t=0}^{T-1} \eta_t \delta_t. \tag{37}$$

For a weighting scheme such that $w_{\max,t} \leq \frac{2}{b}$ so that $\eta_t$ can be fixed to $\eta = \frac{1}{8L\sqrt{T}}$, we obtain

$$f(\bar{\theta}^T) - f(\theta^*) \leq \frac{8L\left\|\theta^0 - \theta^*\right\|^2}{\sqrt{T}} + \frac{1}{T} \sum_{t=0}^{T-1} \delta_t, \tag{38}$$

where $\bar{\theta}^T = \frac{1}{T} \sum_{t=0}^{T-1} \theta^t$.

The traditional convergence bound of minibatch SGD for convex functions can be recovered when $\delta_t = 0$ which occurs when the samples in the batch have uniform weights, $w_{i,t} = \frac{1}{b} \ \forall i \in \mathcal{B}$. We obtain a looser convergence bound when $\delta_t \geq 0$ resulting from the assignment of larger weights to smaller loss gaps. Moreover, down-weighting the importance of samples with low loss gaps leads to a better bound ($\delta_t \leq 0$). Note that the bound in Eq. (32) only holds for reweighting schemes that satisfy $w_{i,t} \leq 2/b$. Thus, enforcing an upper bound on the maximum weight that can be assigned to a single data point. Typically, the minibatch gradient with uniform weights is unbiased, $\mathbb{E}\left[\sum_{i \in \mathcal{B}} \frac{1}{b} \nabla f_i(\theta^t)\right] = \nabla f(\theta^t)$. However, in our setting, the importance weights are functions of the sample functions. Hence, $\mathbb{E}\left[\sum_{i \in \mathcal{B}} w_{i,t} \nabla f_i(\theta^t)\right] \neq \nabla f(\theta^t)$ leading to the analysis of Theorem 2 being more involved. Based on our analysis, we note that the step size $\eta_t$ is a function of the maximum value of the weights, $w_{\max,t} = \frac{2}{b}$ which leads to the choice of $\eta = \frac{1}{8L\sqrt{T}}$ for convergence.

(ii) **Minibatch SGD with Momentum:** Given the loss-based sample reweighting strategies, Eq. (33) characterizes the effects of loss-based reweighting on the convergence bound of reweighted minibatch SGD with momentum when the sample losses $f_i$ are convex. Given the loss-based sample reweighting strategies, we incorporate heavy ball momentum method to improve the convergence result and to characterize the effects of loss-based reweighting on the convergence bound. Hence, we have the following update equations:

$$z_{t+1} = z_t - \eta_t \sum_{i \in \mathcal{B}} w_{i,t} \nabla f_i(\theta_t), \tag{39}$$

$$\theta_{t+1} = \frac{\lambda_{t+1}}{1 + \lambda_{t+1}} \theta_t + \frac{1}{1 + \lambda_{t+1}} z_{t+1}, \tag{40}$$

where $\eta_t$ is the step size and $\eta_t, \lambda_t > 0$. The above iterates $\theta_t$ are equal to the iterates of the following stochastic minibatch heavy ball momentum method Sebbouh et al. (2021).

$$\theta_{t+1} = \theta_t - \frac{\eta_t}{1 + \lambda_{t+1}} \sum_{i \in \mathcal{B}} w_{i,t} \nabla f_i(\theta_t) + \frac{\lambda_t}{1 + \lambda_{t+1}}(\theta_t - \theta_{t-1}).$$

Eq. (33) characterizes the effects of loss-based reweighting on the convergence bound of reweighted minibatch SGD with momentum when the sample losses $f_i$ are convex.

For the minibatch SGD with momentum based on Eq. (40), we have

$$\left\|z^{t+1} - \theta^*\right\|^2 = \left\|z^t - \theta^*\right\|^2 + \left\|z^{t+1} - z^t\right\|^2 + 2\left\langle z^{t+1} - z^t, z^t - \theta^* \right\rangle$$

$$= \left\|z^t - \theta^*\right\|^2 + \left\|\eta_t \sum_{i \in \mathcal{B}} w_{i,t} \nabla f_i(\theta^t)\right\|^2 - 2\left\langle \eta_t \sum_{i \in \mathcal{B}} w_{i,t} \nabla f_i(\theta^t), z^t - \theta^* \right\rangle. \tag{41}$$

Applying expectation on both sides, we get

$$\mathbb{E}\left\|z^{t+1} - \theta^*\right\|^2$$

$$\leq \left\|z^t - \theta^*\right\|^2 + \eta_t^2 \mathbb{E}\left\|\sum_{i \in \mathcal{B}} w_{i,t} \nabla f_i(\theta^t)\right\|^2 - 2\eta_t \mathbb{E}\left[\left\langle \sum_{i \in \mathcal{B}} w_{i,t} \nabla f_i(\theta^t), \theta^t - \theta^* \right\rangle\right]$$

$$- 2\eta_t\lambda_t\mathbb{E}\left[\left\langle\sum_{i\in\mathcal{B}}w_{i,t}\nabla f_i(\theta^t), \theta^t - \theta^{t-1}\right\rangle\right]$$

$$\overset{(a)}{\leq} \|z^t - \theta^*\|^2 + \eta_t^2\mathbb{E}\|\sum_{i\in\mathcal{B}}w_{i,t}\nabla f_i(\theta^t)\|^2 - 2\eta_t\mathbb{E}\left[\sum_{i\in\mathcal{B}}w_{i,t}\left(f_i(\theta^t) - f_i(\theta^*)\right)\right]$$

$$- 2\eta_t\lambda_t\mathbb{E}\left[\sum_{i\in\mathcal{B}}w_{i,t}\left(f_i(\theta^t) - f_i(\theta^{t-1})\right)\right]$$

$$\overset{(b)}{\leq} \|z^t - \theta^*\|^2 + \mathbb{E}\left[\sum_{i\in\mathcal{B}}w_{i,t}(4L\eta_t^2 bw_{i,t} - 2\eta_t)(f_i(\theta^t) - f_i(\theta^*))\right]$$

$$- 2\eta_t\lambda_t\mathbb{E}\left[\sum_{i\in\mathcal{B}}w_{i,t}\left(f_i(\theta^t) - f_i(\theta^{t-1})\right)\right]$$

$$= \|z^t - \theta^*\|^2 + \mathbb{E}\left[\sum_{i\in\mathcal{B}}w_{i,t}(4L\eta_t^2 bw_{i,t} - 2\eta_t - 2\eta_t\lambda_t)(f_i(\theta^t) - f_i(\theta^*))\right]$$

$$+ 2\eta_t\lambda_t\mathbb{E}\left[\sum_{i\in\mathcal{B}}w_{i,t}\left(f_i(\theta^{t-1}) - f_i(\theta^*)\right)\right]$$

$$\overset{(c)}{\leq} \|z^t - \theta^*\|^2 - 2\eta_t\lambda_{t+1}\mathbb{E}\left[\sum_{i\in\mathcal{B}}w_{i,t}(f_i(\theta^t) - f_i(\theta^*))\right] + 2\eta_t\lambda_t\mathbb{E}\left[\sum_{i\in\mathcal{B}}w_{i,t}\left(f_i(\theta^{t-1}) - f_i(\theta^*)\right)\right]$$

$$= \|z^t - \theta^*\|^2 - 2\eta_t\lambda_{t+1}\mathbb{E}\left[\sum_{i\in\mathcal{B}}\frac{1}{b}(f_i(\theta^t) - f_i(\theta^*))\right] + 2\eta_t\lambda_{t+1}\mathbb{E}\left[\sum_{i\in\mathcal{B}}\left(\frac{1}{b} - w_{i,t}\right)(f_i(\theta^t) - f_i(\theta^*))\right]$$

$$+ 2\eta_t\lambda_t\mathbb{E}\left[\sum_{i\in\mathcal{B}}\frac{1}{b}\left(f_i(\theta^{t-1}) - f_i(\theta^*)\right)\right] + 2\eta_t\lambda_t\mathbb{E}\left[\sum_{i\in\mathcal{B}}\left(w_{i,t} - \frac{1}{b}\right)\left(f_i(\theta^{t-1}) - f_i(\theta^*)\right)\right]$$

$$= \|z^t - \theta^*\|^2 - 2\eta_t\lambda_{t+1}\mathbb{E}(f(\theta^t) - f(\theta^*)) + 2\eta_t\lambda_{t+1}\mathbb{E}\left[\sum_{i\in\mathcal{B}}\left(\frac{1}{b} - w_{i,t}\right)(f_i(\theta^t) - f_i(\theta^*))\right]$$

$$+ 2\eta_t\lambda_t\mathbb{E}\left(f(\theta^{t-1}) - f(\theta^*)\right) + 2\eta_t\lambda_t\mathbb{E}\left[\sum_{i\in\mathcal{B}}\left(w_{i,t} - \frac{1}{b}\right)\left(f_i(\theta^{t-1}) - f_i(\theta^*)\right)\right]$$

$$= \|z^t - \theta^*\|^2 - 2\eta_t\lambda_{t+1}\mathbb{E}(f(\theta^t) - f(\theta^*)) + \eta_t\mathbb{E}\left[\sum_{i\in\mathcal{B}}\left(\frac{1}{b} - w_{i,t}\right)(f_i(\theta^t) - f_i(\theta^*))\right]$$

$$+ 2\eta_t\lambda_t\mathbb{E}\left(f(\theta^{t-1}) - f(\theta^*)\right) + 2\eta_t\lambda_t\mathbb{E}\left[\sum_{i\in\mathcal{B}}\left(\frac{1}{b} - w_{i,t}\right)\left(f_i(\theta^t) - f_i(\theta^{t-1})\right)\right] \tag{42}$$

where (a) follows from applying Assumption 1 to the third and fourth terms, (b) follows from applying Lemma 1 to the second term, and (c) follows from setting $\eta_t$ as $\eta_t \leq \frac{1}{4Lbw_{\max,t}}$ and $\lambda_{t+1} = \lambda_t + \frac{1}{2}$. Let $\delta_t = \mathbb{E}\left[\sum_{i\in\mathcal{B}}\left(\frac{1}{b} - w_{i,t}\right)(f_i(\theta^t) - f_i(\theta^*))\right]$, $\mu_t = \mathbb{E}\left[\sum_{i\in\mathcal{B}}\left(\frac{1}{b} - w_{i,t}\right)(f_i(\theta^t) - f_i(\theta^{t-1}))\right]$ and $\sum_{i\in\mathcal{B}}w_{i,t}^2 \leq 1$. Hence, telescoping Eq. (42) from $t = 0$ to $T - 1$ and noting that $z_0 = x_0$ yields

$$\mathbb{E}\|z^T - \theta^*\|^2 \leq \|\theta^0 - \theta^*\|^2 - 2\eta\lambda_{T+1}\mathbb{E}\left(f(\theta^T) - f(\theta^*)\right) + \sum_{t=0}^{T-1}2\eta\delta_t + \sum_{t=0}^{T-1}2\eta\lambda_t\mu_t. \tag{43}$$

With $\lambda_t = t/2$, for a weighting scheme such that $w_{\max,t} \leq \frac{2}{b}$ so that $\eta_t$ can be fixed to $\eta = \frac{1}{8L\sqrt{T+1}}$, we obtain

$$\mathbb{E}[f(\theta^T) - f(\theta^*)] \leq \frac{8L\|\theta^0 - \theta^*\|^2}{\sqrt{T+1}} + \frac{2}{T+1}\sum_{t=0}^{T-1}\delta_t + \frac{2}{T+1}\sum_{t=0}^{T-1}\lambda_t\mu_t. \tag{44}$$

$\square$

## C.4 Non-convex Minibatch SGD

**Non-convex:** Unlike the convex case, the importance weight $w_{i,t}$ is set based on the gradient norm instead of the loss gap $f_i(\theta^t) - f_i(\theta^*)$ when the sample functions $f_i$ are non-convex. Theorem 4 underlines the influence of these importance weights on the convergence bound for the non-convex case. Note that computing the gradient norm requires extra computations, whereas the loss values are readily available in practice.

**Assumption 4.** *The norm of the difference between the sample gradient $\nabla f_i(\theta_t)$ and full gradient $\nabla f(\theta_t)$ is bounded by $\|\nabla f_i(\theta_t) - \nabla f(\theta_t)\|^2 \leq \mathcal{V}^2, \forall \theta_t, i \in [M]$.*

**Theorem 4.** *[Non-convex Minibatch SGD] Given that the objective functions $f_i, i \in [n]$ satisfy Assumption 2 and Assumption 4. Consider a minibatch of data points $|\mathcal{B}| = b$ with reweighting scheme $w_{i,t}$, if $i \in \mathcal{B}$ and 0 otherwise where $\max_{i \in \mathcal{B}} w_{i,t} \leq 2/b$. Then, for $\eta = \frac{1}{8L\sqrt{T}}$, we have*

$$\frac{1}{T} \sum_{t=0}^{T-1} \mathbb{E}\|\nabla f(\theta^t)\|^2 \leq \frac{16L(f(\theta^0) - f^*)}{\sqrt{T}} + \frac{8}{T} \sum_{t=0}^{T-1} \mathbb{E}\left[ \sum_{i \in \mathcal{B}} \left(\frac{1}{b} - w_{it}\right) \|\nabla f_i(\theta^t)\|^2 \right] + \frac{b\mathcal{V}^2}{4\sqrt{T}}.$$
(45)

*Proof.* Applying $L$-smoothness condition

$f(\theta^{t+1})$

$$\leq f(\theta^t) + \langle \nabla f(\theta^t), \theta^{t+1} - \theta^t \rangle + \frac{L}{2}\|\theta^{t+1} - \theta^t\|^2$$

$$= f(\theta^t) - \eta \left\langle \nabla f(\theta^t), \sum_{i \in \mathcal{B}} w_{it} \nabla f_i(\theta^t) \right\rangle + \frac{L\eta^2}{2} \left\| \sum_{i \in \mathcal{B}} w_{it} \nabla f_i(\theta^t) \right\|^2$$

$$= f(\theta^t) - \eta \left\langle \nabla f(\theta^t), \sum_{i \in \mathcal{B}} \frac{1}{b} \nabla f_i(\theta^t) \right\rangle + \eta \left\langle \nabla f(\theta^t), \sum_{i \in \mathcal{B}} \left(\frac{1}{b} - w_{it}\right) \nabla f_i(\theta^t) \right\rangle + \frac{L\eta^2}{2} \left\| \sum_{i \in \mathcal{B}} w_{it} \nabla f_i(\theta^t) \right\|^2$$

$$= f(\theta^t) - \eta \left\langle \nabla f(\theta^t), \sum_{i \in \mathcal{B}} \frac{1}{b} \nabla f_i(\theta^t) \right\rangle + \eta \sum_{i \in \mathcal{B}} \left(\frac{1}{b} - w_{it}\right) \langle \nabla f(\theta^t), \nabla f_i(\theta^t) \rangle + \frac{L\eta^2}{2} \left\| \sum_{i \in \mathcal{B}} w_{it} \nabla f_i(\theta^t) \right\|^2$$

$$\leq f(\theta^t) - \eta \left\langle \nabla f(\theta^t), \sum_{i \in \mathcal{B}} \frac{1}{b} \nabla f_i(\theta^t) \right\rangle + 2\eta \sum_{i \in \mathcal{B}} \left(\frac{1}{b} - w_{it}\right) \|\nabla f(\theta^t)\|^2 + 2\eta \sum_{i \in \mathcal{B}} \left(\frac{1}{b} - w_{it}\right) \|\nabla f_i(\theta^t)\|^2$$

$$+ \frac{L\eta^2}{2} \left\| \sum_{i \in \mathcal{B}} w_{it} \nabla f_i(\theta^t) \right\|^2$$

$$= f(\theta^t) - \eta \left\langle \nabla f(\theta^t), \sum_{i \in \mathcal{B}} \frac{1}{b} \nabla f_i(\theta^t) \right\rangle + 2\eta \sum_{i \in \mathcal{B}} \left(\frac{1}{b} - w_{it}\right) \|\nabla f_i(\theta^t)\|^2 + \frac{L\eta^2}{2} \left\| \sum_{i \in \mathcal{B}} w_{it} \nabla f_i(\theta^t) \right\|^2$$

$$\leq f(\theta^t) - \eta \left\langle \nabla f(\theta^t), \sum_{i \in \mathcal{B}} \frac{1}{b} \nabla f_i(\theta^t) \right\rangle + 2\eta \sum_{i \in \mathcal{B}} \left(\frac{1}{b} - w_{it}\right) \|\nabla f_i(\theta^t)\|^2 + \frac{Lb\eta^2}{2} \sum_{i \in \mathcal{B}} w_{it}^2 \|\nabla f_i(\theta^t)\|^2$$

$$\leq f(\theta^t) - \eta \left\langle \nabla f(\theta^t), \sum_{i \in \mathcal{B}} \frac{1}{b} \nabla f_i(\theta^t) \right\rangle + 2\eta \sum_{i \in \mathcal{B}} \left(\frac{1}{b} - w_{it}\right) \|\nabla f_i(\theta^t)\|^2$$

$$+ Lb\eta^2 \sum_{i \in \mathcal{B}} w_{it}^2 \|\nabla f_i(\theta^t) - \nabla f_i(\theta^t)\|^2 + Lb\eta^2 \sum_{i \in \mathcal{B}} w_{it}^2 \|\nabla f(\theta^t)\|^2$$

$$\leq f(\theta^t) - \eta \left\langle \nabla f(\theta^t), \sum_{i \in \mathcal{B}} \frac{1}{b} \nabla f_i(\theta^t) \right\rangle + 2\eta \sum_{i \in \mathcal{B}} \left(\frac{1}{b} - w_{it}\right) \|\nabla f_i(\theta^t)\|^2$$

$$+ Lb\eta^2 \mathcal{V}^2 + Lb\eta^2 \sum_{i \in \mathcal{B}} w_{i,t}^2 \|\nabla f(\theta^t)\|^2$$

$$(46)$$

where we have used the bound $\left\|\nabla f_i(\theta^t) - \nabla f(\theta^t)\right\|^2 \leq \mathcal{V}^2$ and $\sum_{i \in \mathcal{B}} w_{i,t}^2 \leq 1$. Applying expectation on both sides, we get

$$
\begin{aligned}
\mathbb{E}f(\theta^{t+1}) &\leq f(\theta^t) - \eta \mathbb{E}\left\|\nabla f(\theta^t)\right\|^2 + (2\eta)\mathbb{E}\left[\sum_{i \in \mathcal{B}}\left(\frac{1}{b} - w_{it}\right)\left\|\nabla f_i(\theta^t)\right\|^2\right] \\
&\quad + Lb\eta^2\mathcal{V}^2 + Lb\eta^2\sum_{i \in \mathcal{B}}w_{it}^2\mathbb{E}\left\|\nabla f(\theta^t)\right\|^2 \\
&\leq f(\theta^t) - \eta\mathbb{E}\left\|\nabla f(\theta^t)\right\|^2 + (2\eta)\mathbb{E}\left[\sum_{i \in \mathcal{B}}\left(\frac{1}{b} - w_{it}\right)\left\|\nabla f_i(\theta^t)\right\|^2\right] \\
&\quad + Lb\eta^2\mathcal{V}^2 + Lb^2\eta^2 w_{\max,t}^2\mathbb{E}\left\|\nabla f(\theta^t)\right\|^2 \\
&\leq f(\theta^t) - \frac{\eta}{2}\mathbb{E}\left\|\nabla f(\theta^t)\right\|^2 + 2\eta\mathbb{E}\left[\sum_{i \in \mathcal{B}}\left(\frac{1}{b} - w_{it}\right)\left\|\nabla f_i(\theta^t)\right\|^2\right] + Lb\eta^2\mathcal{V}^2 \quad (47)
\end{aligned}
$$

where $\eta \leq \frac{1}{2Lb^2 w_{\max,t}^2}$ in the last inequality. Telescoping and rearranging the terms yields

$$
\frac{1}{T}\sum_{t=0}^{T-1}\mathbb{E}\|\nabla f(\theta^t)\|^2 \leq \frac{2(f(\theta^0) - f^*)}{\eta T} + \frac{4}{T}\sum_{t=0}^{T-1}\mathbb{E}\left[\sum_{i \in \mathcal{B}}\left(\frac{1}{b} - w_{it}\right)\|\nabla f_i(\theta^t)\|^2\right] + 2Lb\eta\mathcal{V}^2.
$$

$$(48)$$

For a weighting scheme such that $w_{\max,t} \leq \frac{2}{b}$ so that $\eta_t$ can be fixed to $\eta = \frac{1}{8L\sqrt{T}}$, we get

$$
\frac{1}{T}\sum_{t=0}^{T-1}\mathbb{E}\|\nabla f(\theta^t)\|^2 \leq \frac{16L(f(\theta^0) - f^*)}{\sqrt{T}} + \frac{4}{T}\sum_{t=0}^{T-1}\mathbb{E}\left[\sum_{i \in \mathcal{B}}\left(\frac{1}{b} - w_{it}\right)\|\nabla f_i(\theta^t)\|^2\right] + \frac{b\mathcal{V}^2}{4\sqrt{T}}.
$$

$$(49)$$

$\square$

From Theorem 4, we can obtain the convergence bound of traditional minibatch SGD for non-convex functions when $w_{i,t} = \frac{1}{b}\ \forall i \in \mathcal{B}$. Moreover, a looser convergence bound is obtained when $\mathbb{E}\left[\sum_{i \in \mathcal{B}}\left(\frac{1}{b} - w_{it}\right)\|\nabla f_i(\theta^t)\|^2\right] \geq 0$, for $w_{i,t} \leq \frac{1}{b}$.

