# OpenReview forum: "Dynamic Loss-Based Sample Reweighting for Improved Large Language Model Pretraining"
_ICLR.cc/2025/Conference — ICLR 2025 Poster_

### Official Review · Reviewer_9BZv · 2024-11-01

**Soundness:** 3
**Presentation:** 4
**Contribution:** 3
**Rating:** 6
**Confidence:** 3

**Summary:**

The paper presents a framework for dynamically reweighting training samples in large language model (LLM) pretraining based on loss values. This approach contrasts with traditional uniform sampling by adjusting sample importance during training, allowing the model to focus on more informative data and deprioritize redundant samples. The authors introduce three reweighting strategies, each designed to emphasize specific portions of the loss distribution, and establish a theoretical foundation for their effectiveness on model convergence. Experimental results show that these methods lead to faster convergence and improved performance on various benchmarks, especially when combined with existing domain-level reweighting methods like DoGE and DoReMi.

**Strengths:**

- Originality: The paper introduces a novel approach to LLM pretraining by leveraging instance-level, dynamic reweighting based on loss values. Also, the theoretical results justify the methods.

- Quality: The paper combines theoretical analysis with empirical results across multiple benchmarks. The authors provide convergence bounds and use diverse model scales to demonstrate the generalization and scalability.

- Clarity: The methodology and notations are well-defined, with clear mathematical formulation and supporting visualizations.

- Significance: The paper’s instance-level reweighting method enhances training efficiency and model performance. Its compatibility with existing domain-level techniques (e.g., DoGE, DoReMi) highlights its practical value

**Weaknesses:**

1. **Clarity**: The statement “where $L>0$ is the smoothness parameter” under Equation (7) is unclear since $L$ does not show up in Equation (7) and could benefit from further explanation.
2. **Benchmark Choice**: The choice of benchmarks (LogiQA, LogiQA 2, SciQA, PiQA) is not well-justified. Testing on additional benchmarks like MMLU, ARC, hellaswag, boolq and TruthfulQA could strengthen the few-shot evaluation.
3. **Baseline Comparisons**: Table 1 lacks a direct comparison with baseline methods (DoGE and DoReMi), which would provide valuable context for the effectiveness of the proposed approach.
4. **Theoretical Assumptions**: The assumption that the maximum weight is capped at $2/M$ may be too strict, particularly in data selection scenarios with fewer than half the dataset selected. Justification for this assumption's practicality is needed.
5. **Efficiency Discussion**: The paper could better address the efficiency benefits of the proposed methods compared to DoGE and DoReMi, especially regarding computation and implementation costs.

**Questions:**

1. **Benchmark Justification**: Could you elaborate on the selection of LogiQA, LogiQA 2, SciQA, and PiQA as benchmarks? Testing additional benchmarks, such as MMLU, ARC, hellaswag, boolq, and TruthfulQA, could further strengthen the few-shot performance evaluation.

2. **Baseline Comparisons**: Including DoGE and DoReMi as baselines in Table 1 would provide useful context on the effectiveness of the proposed approach. Would you consider adding this comparison?

3. **Justification for $2/M$**: The theoretical assumption cap on maximum weight at $2/M$ seems restrictive. Could you provide further justification or discuss how this assumption holds in practical applications?

4. **Efficiency Discussion**: A deeper comparison of computational and implementation efficiency relative to DoGE and DoReMi could underscore the practical advantages of your method. Could you expand on this?

---

> ### Author Response · Authors · 2024-11-23
>
> Thank you for your thorough reviews and constructive comments. We provide our response to your comments below. If our response resolves your concerns, we would greatly appreciate it if you could consider increasing your score.
>
> **Q1:** The statement “where $L>0$ is the smoothness parameter” under Equation (7) is unclear since $L>0$ does not show up in Equation (7) and could benefit from further explanation.
>
> **A1:** Thanks for noticing this! This was a typo and we have removed it in our revision.
>
> **Q2:**  Could you elaborate on the selection of LogiQA, LogiQA 2, SciQA, and PiQA as benchmarks? Testing additional benchmarks could further strengthen the few-shot performance evaluation.
>
> **A2:** We thank the reviewer for the valuable feedback! For the GPT-2 models, we find that only question-answering tasks work well at this scale, which is why we report 5-shot results on LogiQA, LogiQA-2, SciQ, and PiQA. All other benchmarks we explored yielded random results for all compared algorithms, i.e., they were not useful for model evaluation. The reason is the limited capacity of the GPT-2 models.
>
> To address the reviewer’s concern, **we have conducted new experiments to train additional 1.4B (billion) and 7B model with Llama architecture** on randomly sampled subsets of the FineWeb 15T dataset. For the 1.4B and 7B models, we use 100B and 175B randomly sampled tokens from the FineWeb 15T dataset, respectively. We train the models using the Uniform baseline with uniformly averaged losses and our **LinUpper** strategy.
>
> For both models, we include benchmark results from 13 language understanding & reasoning (LUR) tasks and 6 question-answering (QA) tasks. Please **see the new section 6.4 in our main paper and Appendix A for more details, including hyperparameters and model architectural details**. We report the benchmark results on the last checkpoints.
>
> After fully training the 1.4B model on 100B tokens, we observe a performance gain on 6 out of 13 LUR tasks with an average performance gain of 0.66%. For QA benchmarks, we improved on 5 out of 6 tasks with an average of 1.72%.
> For the 1.4B model, we provide a full overview below and in our updated paper in Appendix A, Tables 8 and 9. The first table shows the language understanding and reasoning benchmarks; the second question-answering tasks.
>
> | Benchmark Name | Uniform | LinUpper (ours) | Difference |
> |:---|---:|---:|---:|
> | ARC Challenge | 33.19 | 33.02 | -0.17 |
> | ARC Easy | 63.80 | 63.38 | -0.42 |
> | COPA | 77.0 | 77.0 | 0.00 |
> | Lambada (OpenAI) | 49.87 | 49.60 | -0.27 |
> | Lambada (Standard) | 45.20 | 43.30 | -1.90 |
> | MMLU | 26.04 | 25.49 | -0.54 |
> | MNLI | 31.91 | 36.92 | 5.01 |
> | MNLI (mismatch) | 32.07 | 36.56 | 4.50 |
> | RTE | 51.26 | 52.35 | 1.08 |
> | SST-2 | 66.86 | 59.40 | -7.45 |
> | TinyARC (norm.) | 40.46 | 40.85 | 0.39 |
> | TinyWinoGrande | 56.17 | 62.11 | 5.93 |
> | WinoGrande | 56.75 | 59.12 | 2.37 |
> | *Mean* | *48.51* | *49.16* | *0.66* |
>
>
>
> | **Benchmark Name** | **Uniform** | **LinUpper (ours)** | **Difference** |
> |:---|---:|---:|---:|
> | BoolQ | 61.59 | 65.47 | 3.88 |
> | LogiQA | 20.28 | 23.66 | 3.38 |
> | LogiQA2 | 26.02 | 27.48 | 1.46 |
> | SciQ | 91.30 | 91.90 | 0.60 |
> | SocialIQA | 43.86 | 45.19 | 1.33 |
> | TriviaQA (Exact Match) | 11.05 | 10.73 | -0.32 |
> | _Mean_ | _42.35_ | _44.07_ | _1.72_ |
>
> **Update**: We have completed the training of the 7B model and now report the final benchmark accuracies after 175B tokens. The training took a total of 9800 GPU hours.
> Overall, our **LinUpper** strategy performs better on 9 out of 13 LUR tasks, with an average improvement across all benchmarks of 1.45%.
> Also, we observe an average performance gain on QA benchmarks of 1.16%. We improve on 5/6 QA tasks. We have updated our paper with these final results.
> The full overview is available in Appendix A, Tables 10 and 11. Below, the first table shows the language understanding and reasoning benchmarks; the second question-answering tasks.
>
> | **Benchmark Name** | **Uniform** | **LinUpper (ours)** | **Difference** |
> |:-------------------|--------:|-------------------:|-----------:|
> | ARC Challenge | 41.81 | 39.42 | -2.39 |
> | ARC Easy | 72.77 | 74.24 | 1.47 |
> | COPA | 83.00 | 84.00 | 1.00 |
> | Lambada (OpenAI) | 61.28 | 62.43 | 1.14 |
> | Lambada (Standard) | 58.08 | 58.80 | 0.72 |
> | MMLU | 24.33 | 25.81 | 1.47 |
> | MNLI | 32.28 | 34.24 | 1.97 |
> | MNLI (mismatch) | 32.35 | 34.01 | 1.66 |
> | RTE | 50.54 | 49.10 | -1.44 |
> | SST-2 | 49.89 | 69.38 | 19.50 |
> | TinyARC | 36.17 | 32.80 | -3.37 |
> | TinyWinoGrande | 49.09 | 44.20 | -4.88 |
> | WinoGrande | 65.43 | 67.32 | 1.89 |
> | _Mean_ | _50.54_ | _51.98_ | _1.45_ |
>
> | **Benchmark Name** | **Uniform** | **LinUpper (ours)** | **Difference**
> |:---|---:|---:|---:
> | BoolQ | 73.52 | 74.4 | 0.89
> | LogiQA (norm.) | 23.35 | 27.65 | 4.30
> | LogiQA2 (norm.) | 26.72 | 27.86 | 1.15
> | SciQ (norm.) | 95.20 | 95.00 | -0.20
> | SocialIQA | 47.80 | 48.11 | 0.31
> | TriviaQA (Exact Match) | 22.30 | 22.79 | 0.49
> | _Mean_ | _48.15_ | _49.3_ | _1.16_

---

> ### Author Response · Authors · 2024-11-23
>
> ...continue **A2**.
>
> These improvements are particularly noteworthy given the scale of these LLama models, highlighting the scalability and effectiveness of our method.
>
> **Q3:** Including DoGE and DoReMi as baselines in Table 1 would provide useful context on the effectiveness of the proposed approach. Would you consider adding this comparison?
>
> **A3:** Thanks for the question! Note that the methods DoGE and DoReMi are orthogonal to the uniform domain sampling approaches in Table 1, i.e., all methods in this table can also be run with DoGE or DoReMi. This is why we organized our experiments into two different subsections, one that compares algorithms under uniform domain sampling setting, and a second subsection that deals with the non-uniform domain sampling setup (in which we use DoGE and DoReMi as the optimal domain sampling estimators).
>
> **Q4:** The theoretical assumption cap on maximum weight at $\frac{2}{M}$ seems restrictive. Could you provide further justification or discuss how this assumption holds in practical applications?
>
> **A4:** Thanks for the comment! Note that the inequality $w_i \leq \frac{2}{M}$ is not an assumption but rather a finding of our theoretical analysis (Proposition 1) that supports the benefits of some reweighting strategies such as LinUpper. The only assumptions required for Theorem 1 to hold are the widely adopted Assumptions 1, 2, and 3.
>
> **Q5:** A deeper comparison of computational and implementation efficiency relative to DoGE and DoReMi could underscore the practical advantages of your method. Could you expand on this?
>
> **A5:** Thank you for the insightful question! Our reweighting approach is designed with efficiency at its core, minimizing computational overhead associated with the sample weights computation by innovatively leveraging information from sample loss values. In particular, compared to DoGE and DoReMi, which train additional auxiliary models (**DoReMi trains 2 additional separate models and DoGE trains 1 additional proxy model**), our approach does not require training any additional auxiliary model for computing the instance weights. Hence, in terms of resource consumption, our method requires about 3 times less compute than DoReMi and 2 times less compute than DoGE. Furthermore, in terms of implementation our method enjoys huge simplicity compared to DoGE and DoReMi, which implement additional bilevel/minimax based optimization loops for training the auxiliary models. Instead, our Algorithm 1 can be seamlessly integrated into existing training pipelines for LLMs. To demonstrate how our method can be easily integrated into a traditional multi-GPU training script, we have added a new section in the appendix (Appendix A) that provides a practical implementation in Pytorch. We hope that this further clarifies some details about our method.

---

> > ### Comment · Reviewer_9BZv · 2024-11-27
> >
> > Thanks for the detailed answer and additional experiments. Most of my concerns have been addressed. Still, I believe $w_i \leq \frac{2}{M}$ lies the foundation of the theoretical justification of the proposed method. And this cap on maximum weight is too much restrictive for a data reweighting method. Also, I don't see how this holds in practical applications. Therefore, I will keep my score unchanged.

---

> ### Author Response · Authors · 2024-11-27
>
> Dear Reviewer 9BZv,
>
> Many thanks for your reply. We further clarify the meaning of our theoretical upper bound on the weights as follows.
>
>
> First, we note that in the case of SGD with a minibatch size of b, the *uniformly weighted* SGD (i.e., without reweighting) has a weight of 1/b for each sample in the minibatch, since the weights of all the samples in the minibatch sum to one. In the case of *reweighting*, our theoretical upper bound is that any weight is upper bounded by 2/b. Therefore, this weight upper bound simply **means that the weight of any sample (after reweighting) should not be more than twice the uniform weight.** Intuitively, the reason of having such an upper bound is that, if the weights of some samples are very large, some other samples would have very small weights, causing an extremely skewed data distribution that is not helpful for training. Instead, if the samples are weighted properly within a reasonable range, emphasizing more useful samples but not causing extreme skewness, it is helpful for training. We also note that Theorem 1 in the paper is a special case of Theorem 2 with b=M, which corresponds to the case of full gradient descent and we use it as a theoretical foundation to motivate our reweighting approach. Theorem 2 is more general and aligns with the practical setting when SGD is used.
>
>
> We would also like to explain that our LinUpper reweighting method indeed satisfies the upper bound of $w_i < 2/b$ and any strategy that assigns the same weight to half of the data points and smaller weights to the other half (such as LinUpper) will satisfy the upper bound of $w_i < \frac{2}{b}$. The proof is straightforward when noting that the sum of all weights needs to be 1.
>
> To demonstrate that in practice the upper bound of $w_i < 2/b$ is in fact satisfied by our LinUpper method, we provide the distribution of maximum weights (please see Figure 6 in Appendix A.5) during the pretraining of the 7B parameter model with our LinUpper method, which is the new experiment that we added in the revision. In this experiment, the minibatch size is $128$ and hence the upper bound is $\frac{2}{128} = 0.0156$. As you can see, the maximum weight (which is around $0.01$) by our LinUpper method is clearly below the upper bound of $0.0156$. The fact that we obtain better performance when training using our LinUpper method compared to the uniform weighting baseline confirms the usefulness of reweighting within this upper bound.
>
>
> We also note that we have updated the results of the 7B parameter model training (in both the paper and the previous comments) that we obtained after a full run with 175B tokens of training data.
>
> We hope that this addresses your concern. Please let us know if you have further questions.
>
> Regards,
>
> Authors of Paper 13352

---

> > ### Author Response · Authors · 2024-12-02
> >
> > Dear Reviewer 9BZv,
> >
> > Thank you once again for your valuable time and effort in reviewing our paper. As the reviewer-author discussion period ends soon, we would greatly appreciate your feedback on whether our response to your last comment addresses your concern. If you have any further questions, please feel free to let us know, and we will be happy to provide more clarifications.
> >
> > Thank you for your thoughtful review and feedback!
> >
> > Regards,
> >
> > Authors of Paper 13352

---

### Official Review · Reviewer_AvfZ · 2024-11-03

**Soundness:** 3
**Presentation:** 2
**Contribution:** 2
**Rating:** 6
**Confidence:** 2

**Summary:**

The paper introduces novel algorithms for dynamic, instance-level data reweighting aimed at improving the efficiency and effectiveness of large language model (LLM) pretraining. The proposed methods adjust the weight of each training sample based on its loss value in an online fashion, allowing the model to dynamically focus on more informative or important samples during different training stages. The paper also provides a theoretical framework for analyzing the impact of these reweighting strategies on the convergence of gradient-based optimization.

**Strengths:**

(1) The paper introduces an innovative and practical methodology for dynamically reweighting training samples based on their loss values, which could significantly enhance the efficiency and effectiveness of LLM pretraining.

(2) The empirical findings are robust, demonstrating that the proposed reweighting strategies lead to faster convergence and improved performance across a variety of tasks and model scales.

(3) The proposed algorithms can be seamlessly integrated into existing training pipelines with little to no computational overhead, making them highly practical for real-world applications.

**Weaknesses:**

(1) The paper might benefit from additional implementation details to enable reproducibility of the study, especially for the proposed reweighting algorithms.

(2) While the evaluation is comprehensive, further ablation studies could strengthen the paper by isolating the impact of different components of the proposed methods.

(3) It would be helpful to have more detailed discussions comparing the proposed methods with other widely-known reweighting or data selection techniques beyond those already mentioned.

**Questions:**

1. Could you provide more detailed implementation steps for the dynamic reweighting algorithms? Specifically, information on how to normalize the losses, apply the reweighting strategies, and integrate these into existing training pipelines would be beneficial for reproducibility.
﻿
2. The curriculum adjustment mechanism using a temperature parameter is an interesting aspect of your approach. Could you provide more details or results on how different values of the temperature parameter affect the performance and convergence of the model? This would help in understanding the sensitivity and effectiveness of this component.

3. Your theoretical analysis mentions that the reweighting schemes avoid overfitting to outliers. Could you provide empirical evidence or additional discussion on how well your methods handle outliers in practice, especially compared to other robust optimization techniques?
﻿
4. Your results indicate that the benefits of your reweighting strategies are more pronounced in larger models. Could you provide more insights or hypotheses on why this might be the case? Additionally, discussing any observed limitations or challenges when applying these strategies to smaller models would be helpful.

---

> ### Author Response · Authors · 2024-11-23
>
> Thank you for your thorough reviews and constructive comments. We provide our response to your comments below. If our response resolves your concerns, we would greatly appreciate it if you could consider increasing your score.
>
> **Q1:** Could you provide more detailed implementation steps for the dynamic reweighting algorithms? Specifically, information on how to normalize the losses, apply the reweighting strategies, and integrate these into existing training pipelines would be beneficial for reproducibility.
>
> **A1:** Thanks for the question! We provide a detailed explanation of the steps in our dynamic reweighting method (Algorithm 1 in our paper). Additionally, to facilitate reproducibility of our method we have included a practical implementation in Pytorch that demonstrates how our method can be seamlessly integrated into standard multi-GPU training pipelines (please see Appendix A).
>
> **Step 1: Loss Normalization [line 5 in Algorithm 1]**
>
> The normalization step ensures that the losses are scaled consistently for reweighting. Given a batch of losses $\{f_{i,t}\}, i=1,...,b$, where $b$ is the batchsize, compute:
>
> $h_{i,t} = \frac{2(f_{i,t} - f_{\text{min}})}{f_{\text{max}} - f_{\text{min}}} - 1$.
>
> Here, $f_\text{min}$ and $f_\text{max}$​ are the minimum and maximum loss values in the current batch. This linear transformation maps the losses to the interval [−1,1], ensuring consistent scaling across different training batches. This allows the reweighting functions to operate on a consistent and bounded range of values.
> The normalization step outputs the normalized losses $h_{i,t}, i=1,...,b$ to be used in the next step.
>
> **Step 2: Applying Reweighting Strategies  [line 6 in Algorithm 1]**
>
> Once the losses are normalized, the chosen reweighting strategy is applied:
>
> LinUpper: This strategy downweights the low-loss samples and caps the weight applied to high-loss samples, ensuring that outliers do not dominate the training process.
>
> Quadratic: This strategy emphasizes the medium-loss samples by penalizing both low- and high-loss values.
>
> Extremes: This strategy assigns higher importance to samples with extreme loss values, whether high or low.
>
> This step outputs the strategic weights $s_{i,t},  i=1,...,b$ to be adjusted by the next curriculum step.
>
> **Step 3: Curriculum Adjustment  [line 7 in Algorithm 1]**
>
> One we obtain the strategic weights $s_{i,t},  i=1,...,b$, apply the softmax-based adjustment controlled by the temperature parameter $r$​:
>
> $w_{i,t} = \frac{\exp(s_{i,t} / r_t)}{\sum_{j=1}^b \exp(s_{j,t} / r_t)}$
>
> Start with a higher $r_t$​ (e.g., $r_0$​), ensuring more uniform weights early in training. Reduce $r$ as training progresses to allow for sharper differentiation in sample importance.
>
> **Step 4: Gradient-Based Parameter Update [line 8 in Algorithm 1]**
>
> After computing the sample weights $\{w_{i,t}\}, i=1,...,b$​, update the model parameters using the weighted gradient:
>
> $\theta^{t+1} = \theta^t - \eta \sum_{i=1}^b w_{i,t} \nabla f_{i,t}$
>
> This ensures that each sample contributes to the parameter update proportional to its dynamically associated weight.
>
> Please see Appendix A for the Python code for each of these steps and a practical integration into existing training pipelines.
>
>
> **Q2:** The curriculum adjustment mechanism using a temperature parameter is an interesting aspect of your approach. Could you provide more details or results on how different values of the temperature parameter affect the performance and convergence of the model? This would help in understanding the sensitivity and effectiveness of this component.
>
> **A2:** Great question! We have conducted additional experiments using different values of $r$ to understand the sensitivity of our methods to the value of $r$. The perplexity plots and benchmark results can be found in Appendix A.4. Figure 5 in Appendix A.4 shows that when the value of r is large (e.g. r=1), the performance of our method becomes closer to that of the uniform baseline. Decreasing the value of r leads to diminished effect of low-loss samples, but this can also have a negative effect on the performance when r is too small (eg. r=0.2), potentially due to data wastage by overfiltering the low-loss samples.

---

> ### Author Response · Authors · 2024-11-23
>
> **Q3:** Your theoretical analysis mentions that the reweighting schemes avoid overfitting to outliers. Could you provide empirical evidence or additional discussion on how well your methods handle outliers in practice, especially compared to other robust optimization techniques?
>
> **A3:** Thank you for your insightful question! Unlike traditional robust optimization methods, such as those based on distributionally robust optimization (DRO), our methods impose a cap on sample weights ($w_i \leq 2/M$), as justified by our theoretical analysis (Proposition 1). This cap prevents over-reliance on a small subset of high-loss samples, which can lead to overfitting to outliers. In contrast, DRO methods often focus heavily on worst-case scenarios, potentially leading to suboptimal generalization when the training data contains a substantial amount of noisy samples.
> To demonstrate this benefit of our method in practice, we conducted new experiments to compare with the traditional KL-divergence regularized distributionally robust optimization (DRO-KL). First, we created a synthetic regression problem (similar to the regression problem in Appendix B) with 25% of the data being outliers to compare the robustness of the different methods against outliers.
>
> The experiment's plots can be found in Figure 7 of Appendix B.3. We note the following: The DRO-KL approach because it focuses heavily on the hard samples, it diverges when we use the same learning rate as the other 2 methods (our LinUpper & the Uniform baseline). The DRO-KL method required a smaller learning rate to converge, in which case it converged slower than our method.
> Furthermore, we have also included the DRO-KL method among the compared baselines for the GPT-2-medium experiments. On average, our LinUpper method outperforms the DRO-KL method, which we attribute to its ability to handle outliers (see Table 12 in Appendix B.3 for the detailed results).
>
> **Q4:** Your results indicate that the benefits of your reweighting strategies are more pronounced in larger models. Could you provide more insights or hypotheses on why this might be the case? Additionally, discussing any observed limitations or challenges when applying these strategies to smaller models would be helpful.
>
> **A4:** Thanks for the question! Indeed, our results show that the benefits of our reweighting strategies become more pronounced as model size increases. Below, we provide further insights and hypotheses to explain this observation, as well as discuss potential limitations and challenges when applying our methods to smaller models.
>
> **(a) Insights for Larger Models**
>
> Larger models have a higher capacity to learn from both the clean (informative) and noisy data. As a result, they are more sensitive to the training dynamics and can better leverage reweighting strategies that emphasize certain subsets of the data. Specifically:
>
> Overfitting Control: Larger models are inherently more prone to overfitting due to their increased parameter space. Our reweighting schemes, particularly those that down-weight low-loss samples (e.g. **LinUpper**), help mitigate this risk by preventing the model from overemphasizing redundant or easy examples.
>
> Representation Capacity: Larger models can represent complex patterns in the data more effectively. This allows them to benefit more from focusing on medium- or high-loss samples, which might correspond to harder but useful examples that smaller models struggle to learn.
>
> **(b) Hypotheses for Smaller Models**
>
> In contrast, smaller models often lack the capacity to learn complex patterns effectively and tend to underfit. Hence, since these limited-capacity models are less capable of exploiting harder examples, the benefits of emphasizing medium- or high-loss samples are diminished. These models may even rely more on easy examples to make incremental improvements, which reduces the effectiveness of reweighting strategies like **LinUpper** or **Quadratic**.
>
> **(c) Potential limitations for Smaller Models**
>
> Intuitively, in some cases reweighting strategies like **LinUpper** could even introduce instability for smaller models by shifting focus away from easy-to-learn samples, which are crucial for their limited capacity. However, in practice, as validated by our various experiments, our reweighting method **LinUpper** provides a consistent performance boost for larger models, while still achieving competitive performance for smaller models.

---

> ### Author Response · Authors · 2024-11-23
>
> **Note:** We wanted to let the reviewer know that we have conducted extensive additional experiments in which we trained larger 1.4B (billion) and 7B models. For both models, we include benchmark results from 13 language understanding & reasoning (LUR) tasks and 6 question-answering (QA) tasks.
>
> After fully training the 1.4B model on 100B tokens, we observe a performance gain on 11 out of 19 benchmarks with an average performance gain of 0.99%. For the larger 7B parameter model (trained on 175B tokens), we improved performance on 14 out of 19 tasks with an average gain of 1.35%.
>
> Please **see the new section 6.4 in our main paper and Appendix A for more details.**
>
> These improvements are particularly noteworthy given the scale of these LLama models, highlighting the scalability and effectiveness of our method.

---

> ### Author Response · Authors · 2024-11-27
>
> Dear Reviewer AvfZ,
>
> Thank you once again for your valuable time and effort in reviewing our paper. We have responded to your comments in the threads above and also added new experiments to answer your questions. All edits and added parts in our revised paper are in blue.
>
> As the reviewer-author discussion period ends soon, we would greatly appreciate your feedback on whether our responses and updates address your concerns. If you have any further questions, please feel free to let us know, and we will be happy to provide more clarifications. Otherwise, if you find that your concerns have been resolved, we kindly ask you to consider raising your score.
>
> Thank you for your thoughtful review and feedback!
>
> Regards,
>
> Authors of Paper 13352

---

> > ### Author Response · Authors · 2024-11-29
> >
> > Dear Reviewer AvfZ,
> >
> > Thank you once again for your valuable time and effort in reviewing our paper. We have responded to your comments in the threads above and also added new experiments to answer your questions. All edits and added parts in our revised paper are in blue.
> >
> > As the reviewer-author discussion period ends soon, we would greatly appreciate your feedback on whether our responses and updates address your concerns. If you have any further questions, please feel free to let us know, and we will be happy to provide more clarifications. Otherwise, if you find that your concerns have been resolved, we kindly ask you to consider raising your score.
> >
> > Thank you for your thoughtful review and feedback!
> >
> > Regards,
> >
> > Authors of Paper 13352

---

> > > ### Author Response · Authors · 2024-12-01
> > >
> > > Dear Reviewer AvfZ,
> > >
> > > Thank you once again for your valuable time and effort in reviewing our paper. We have responded to your comments in the threads above and also added new experiments to answer your questions. All edits and added parts in our revised paper are in blue.
> > >
> > > As the reviewer-author discussion period will end in two days, we would greatly appreciate your feedback on whether our responses and updates address your concerns. If you have any further questions, please feel free to let us know, and we will be happy to provide more clarifications. Otherwise, if you find that your concerns have been resolved, we kindly ask you to consider raising your score.
> > >
> > > Thank you for your thoughtful review and feedback!
> > >
> > > Regards,
> > >
> > > Authors of Paper 13352

---

> ### Comment · Reviewer_AvfZ · 2024-12-03
> **Official Comment**
>
> Thanks for the detailed response! Most of my concerns have been addressed, and I appreciate the effort you put into clarifying the points I raised. I've increased the score to reflect these improvements. It would be great if the authors could include these improvements in the final version.

---

> > ### Author Response · Authors · 2024-12-03
> >
> > Dear Reviewer AvfZ,
> >
> > We thank the reviewer very much for checking our response and increasing the rating. We will make sure to include all updates in our final version. Many thanks again for your effort into this review process!
> >
> > Regards,
> > Authors of Paper 13352

---

### Official Review · Reviewer_u8Tc · 2024-11-05

**Soundness:** 2
**Presentation:** 2
**Contribution:** 2
**Rating:** 6
**Confidence:** 4

**Summary:**

This paper introduces a dynamic, instance-level data reweighting approach aimed at enhancing the efficiency and effectiveness of large language model (LLM) pretraining. Traditional LLM training paradigms apply uniform sampling to datasets, ignoring individual sample importance. Current reweighting strategies are mostly group-level and lack the adaptability required for instance-specific optimization during training. The authors propose a novel approach that reweights samples dynamically based on their loss values, helping the model to focus on more informative samples. They also provide a new theoretical framework for analyzing how loss-based reweighting impacts convergence rates, demonstrating that deprioritizing low-loss samples leads to faster convergence and improved performance. Empirical results confirm the effectiveness of this approach across various tasks, from large-scale LLM pretraining to smaller problems like linear regression.

**Strengths:**

1. Innovative Instance-Level Reweighting Mechanism: The paper introduces a fine-grained, instance-level reweighting strategy, which addresses the limitations of traditional group-level reweighting methods.

2. Theoretical Framework for Convergence Analysis: The authors provide a novel theoretical framework that explicitly characterizes the effect of loss-based reweighting on convergence bounds. This contribution not only supports the empirical findings but also adds a theoretical foundation that enhances the credibility of the proposed approach.

**Weaknesses:**

1. Although the authors state that the proposed method is designed for LLM pretraining, the evaluation is conducted solely on QA tasks. Generally, QA tasks are simpler than generation or reasoning tasks. Therefore, the benefits of pretraining LLMs may not be adequately assessed under this setting.

2. Experimental settings. In the experimental section, the authors evaluate only on smaller language models, specifically GPT-2 models with fewer than 1 billion parameters. Typically, effective LLM pretraining methods should demonstrate scalability to models with at least 10 billion parameters to ensure practical applicability. Consequently, the practicality of the proposed method is not fully substantiated in this study.

3. There are several typos in this paper. For example, "LANGGUAGE" should be 'Language". "fined-grained" should be "fine-grained". "deprioritize redundant or uninformative data" should be "deprioritizing redundant or uninformative data".

**Questions:**

See weaknesses.

---

> ### Author Response · Authors · 2024-11-23
>
> Thank you for your thorough reviews and constructive comments. We provide our response to your comments below. If our response resolves your concerns, we would greatly appreciate it if you could consider increasing your score.
>
> **Q1:** Although the authors state that the proposed method is designed for LLM pretraining, the evaluation is conducted solely on QA tasks. Generally, QA tasks are simpler than generation or reasoning tasks. Therefore, the benefits of pretraining LLMs may not be adequately assessed under this setting.
>
> **A1:** We thank the reviewer for the valuable suggestions! We have evaluated on more benchmarks, including reasoning and language understanding tasks.
>
> For the GPT-2 models, we find that only question-answering tasks work well at this scale, which is why we report 5-shot results on LogiQA, LogiQA-2, SciQ, and PiQA. All other benchmarks we explored yielded random results for all compared algorithms, i.e., they were not useful for model evaluation. The reason is the limited capacity of the GPT-2 models.
>
> To address the reviewer’s concern, **we have conducted new experiments to train additional 1.4B (billion) and 7B model with Llama architecture** on randomly sampled subsets of the FineWeb 15T dataset. For the 1.4B and 7B models, we use 100B and 175B randomly sampled tokens from the FineWeb 15T dataset, respectively. We train the models using the Uniform baseline with uniformly averaged losses and our **LinUpper** strategy.
>
> For both models, we include benchmark results from 13 language understanding & reasoning (LUR) tasks and 6 question-answering (QA) tasks. Please **see the new section 6.4 in our main paper and Appendix A for more details, including hyperparameters and model architectural details**. We report the benchmark results on the last checkpoints.
>
> After fully training the 1.4B model on 100B tokens, we observe a performance gain on 6 out of 13 LUR tasks with an average performance gain of 0.66%. For QA benchmarks, we improved on 5 out of 6 tasks with an average of 1.72%.
> For the 1.4B model, we provide a full overview below and in our updated paper in Appendix A, Tables 8 and 9. The first table shows the language understanding and reasoning benchmarks; the second question-answering tasks.
>
> | Benchmark Name | Uniform | LinUpper (ours) | Difference |
> |:---|---:|---:|---:|
> | ARC Challenge | 33.19 | 33.02 | -0.17 |
> | ARC Easy | 63.80 | 63.38 | -0.42 |
> | COPA | 77.0 | 77.0 | 0.00 |
> | Lambada (OpenAI) | 49.87 | 49.60 | -0.27 |
> | Lambada (Standard) | 45.20 | 43.30 | -1.90 |
> | MMLU | 26.04 | 25.49 | -0.54 |
> | MNLI | 31.91 | 36.92 | 5.01 |
> | MNLI (mismatch) | 32.07 | 36.56 | 4.50 |
> | RTE | 51.26 | 52.35 | 1.08 |
> | SST-2 | 66.86 | 59.40 | -7.45 |
> | TinyARC (norm.) | 40.46 | 40.85 | 0.39 |
> | TinyWinoGrande | 56.17 | 62.11 | 5.93 |
> | WinoGrande | 56.75 | 59.12 | 2.37 |
> | *Mean* | *48.51* | *49.16* | *0.66* |
>
>
>
> | **Benchmark Name** | **Uniform** | **LinUpper (ours)** | **Difference** |
> |:---|---:|---:|---:|
> | BoolQ | 61.59 | 65.47 | 3.88 |
> | LogiQA | 20.28 | 23.66 | 3.38 |
> | LogiQA2 | 26.02 | 27.48 | 1.46 |
> | SciQ | 91.30 | 91.90 | 0.60 |
> | SocialIQA | 43.86 | 45.19 | 1.33 |
> | TriviaQA (Exact Match) | 11.05 | 10.73 | -0.32 |
> | _Mean_ | _42.35_ | _44.07_ | _1.72_ |
>
> **Update**: We have completed the training of the 7B model and now report the final benchmark accuracies after 175B tokens. The training took a total of 9800 GPU hours.
> Overall, our **LinUpper** strategy performs better on 9 out of 13 LUR tasks, with an average improvement across all benchmarks of 1.45%.
> Also, we observe an average performance gain on QA benchmarks of 1.16%. We improve on 5/6 QA tasks. We have updated our paper with these final results.
> The full overview is available in Appendix A, Tables 10 and 11. Below, the first table shows the language understanding and reasoning benchmarks; the second question-answering tasks.
>
> | **Benchmark Name** | **Uniform** | **LinUpper (ours)** | **Difference** |
> |:-------------------|--------:|-------------------:|-----------:|
> | ARC Challenge | 41.81 | 39.42 | -2.39 |
> | ARC Easy | 72.77 | 74.24 | 1.47 |
> | COPA | 83.00 | 84.00 | 1.00 |
> | Lambada (OpenAI) | 61.28 | 62.43 | 1.14 |
> | Lambada (Standard) | 58.08 | 58.80 | 0.72 |
> | MMLU | 24.33 | 25.81 | 1.47 |
> | MNLI | 32.28 | 34.24 | 1.97 |
> | MNLI (mismatch) | 32.35 | 34.01 | 1.66 |
> | RTE | 50.54 | 49.10 | -1.44 |
> | SST-2 | 49.89 | 69.38 | 19.50 |
> | TinyARC | 36.17 | 32.80 | -3.37 |
> | TinyWinoGrande | 49.09 | 44.20 | -4.88 |
> | WinoGrande | 65.43 | 67.32 | 1.89 |
> | _Mean_ | _50.54_ | _51.98_ | _1.45_ |
>
> | **Benchmark Name** | **Uniform** | **LinUpper (ours)** | **Difference** |
> |:---|---:|---:|---:|
> | BoolQ | 73.52 | 74.4 | 0.89 |
> | LogiQA (norm.) | 23.35 | 27.65 | 4.30 |
> | LogiQA2 (norm.) | 26.72 | 27.86 | 1.15 |
> | SciQ (norm.) | 95.20 | 95.00 | -0.20 |
> | SocialIQA | 47.80 | 48.11 | 0.31 |
> | TriviaQA (Exact Match) | 22.30 | 22.79 | 0.49 |
> | _Mean_ | _48.15_ | _49.3_ | _1.16_ |

---

> ### Author Response · Authors · 2024-11-23
>
> ...continue **A1**.
>
> These improvements are particularly noteworthy given the scale of these LLama models, highlighting the scalability and effectiveness of our method.
>
> **Q2:** Experimental settings. In the experimental section, the authors evaluate only on smaller language models, specifically GPT-2 models with fewer than 1 billion parameters. Typically, effective LLM pretraining methods should demonstrate scalability to models with at least 10 billion parameters to ensure practical applicability. Consequently, the practicality of the proposed method is not fully substantiated in this study.
>
> **A2:** Thanks for the suggestions!  We understand 300M parameter models may not yield a full picture of our reweighting strategy’s effectiveness on significantly larger foundation models. Therefore, **We have conducted new experiments on larger 1.4B and 7B models with Llama architecture**. Please see our response to **Q1** for the experiments’ details and comparisons. We also added these experiments in section 6.4 and in Appendix A of our revision.
>
> We appreciate the reviewer’s suggestion of even larger 10+ billion parameter models, however pretraining from scratch models at this scale requires a significant amount of resources and time (which is typically not available to academic groups). We are exploring possibilities to pretrain at this scale, which could indeed further justify the practicability of our reweighting framework. Additionally, note that the size of 1 billion parameters is widely adopted for conducting ablation studies for LLMs [1]. As the benchmark results in our response to Question 1 reinforce the results we present in the original paper, we hope the increased model size is satisfactory for the reviewer.
>
> **Q3:** There are several typos in this paper. For example, "LANGGUAGE" should be 'Language". "fined-grained" should be "fine-grained". "deprioritize redundant or uninformative data" should be "deprioritizing redundant or uninformative data".
>
> **A3:** We thank the reviewer for noticing these typos! We have corrected these in our revision.
>
> Reference:
>
> [1] Cody Blakeney, et. al. Does your data spark joy? Performance gains from domain upsampling at the end of training, 2024.

---

> > ### Comment · Reviewer_u8Tc · 2024-11-27
> >
> > Thanks for your detailed response. I think with the additional experimental results, the quality of this paper has been improved to meet the ICLR standard.

---

> ### Author Response · Authors · 2024-11-27
>
> Dear Reviewer u8Tc,
>
> We thank the reviewer very much for checking our response and increasing the rating. Many thanks for your effort into this review process!
>
> Regards,
> Authors of Paper 13352

---

> > ### Author Response · Authors · 2024-11-27
> >
> > Dear Reviewer u8Tc,
> >
> > Again, many thanks for your feedback!
> >
> > We have completed the training of the 7B model and now report the final benchmark accuracies after 175B tokens. The training took a total of 9800 GPU hours.
> > Overall, our **LinUpper** strategy performs better on 9 out of 13 LUR tasks, with an average improvement across all benchmarks of 1.45%.
> > Also, we observe an average performance gain on QA benchmarks of 1.16%. We improve on 5 out of 6 QA tasks. We have updated our paper and the previous comments with these final results.
> >
> >
> > Regards,
> >
> > Authors of Paper 13352

---

### Author Response · Authors · 2024-11-25

Dear Reviewers,

Thank you once again for your valuable time and effort in reviewing our paper. We have responded to your comments in the threads below and also added new experiments to answer your questions. All edits and added parts in our revised paper are in blue.

As the reviewer-author discussion period ends in two days, we would greatly appreciate your feedback on whether our responses and updates address your concerns. If you have any further questions, please feel free to let us know, and we will be happy to provide more clarifications. Otherwise, if you find that your concerns have been resolved, we kindly ask you to consider raising your score.

Thank you for your thoughtful review and feedback!

Best regards,
Authors of Paper 13352

---

### Meta-Review · Area_Chair_vBob · 2024-12-20

**Metareview:**

This paper introduces dynamic, instance-level data reweighting algorithms for large language model (LLM) pretraining, which adaptively adjust sample weights based on loss values during training. By focusing on informative samples and deprioritizing redundant data, these methods improve training efficiency and effectiveness. A new theoretical framework analyzes their impact on gradient-based optimization, with empirical results showing faster convergence and better performance across diverse tasks.

The paper though borderline, seems like can be a good addition to ICLR. The reviewers have provided detailed reviews and have brought up several important issues in the paper. I would encourage the authors to please go through the reviews and improve the paper for the next version.

**Additional Comments On Reviewer Discussion:**

The paper though borderline, seems like can be a good addition to ICLR. The reviewers have provided detailed reviews and have brought up several important issues in the paper. I would encourage the authors to please go through the reviews and improve the paper for the next version.

---

### Decision · Program_Chairs · 2025-01-22

Accept (Poster)